# Towards One-shot Neural Combinatorial Solvers: Theoretical and Empirical Notes on the Cardinality-Constrained Case

**Runzhong Wang**[1]**, Li Shen**[2]**, Yiting Chen**[1]**, Xiaokang Yang**[1]**, Dacheng Tao**[2]**, Junchi Yan**[1]*
[1]MoE Key Lab of Artificial Intelligence, Shanghai Jiao Tong University    [2]JD Explore Academy
{runzhong.wang,sjtucyt,xkyang,yanjunchi}@sjtu.edu.cn
{mathshenli,dacheng.tao}@gmail.com
Code: https://github.com/Thinklab-SJTU/One-Shot-Cardinality-NN-Solver

## Abstract

One-shot non-autoregressive neural networks, different from RL-based ones, have been actively adopted for solving combinatorial optimization (CO) problems, which can be trained by the objective score in a self-supervised manner. Such methods have shown their superiority in efficiency (e.g. by parallelization) and potential for tackling predictive CO problems for decision-making under uncertainty. While the discrete constraints often become a bottleneck for gradient-based neural solvers, as currently handled in three typical ways: 1) adding a soft penalty in the objective, where a bounded violation of the constraints cannot be guaranteed, being critical to many constraint-sensitive scenarios; 2) perturbing the input to generate an approximate gradient in a black-box manner, though the constraints are exactly obeyed while the approximate gradients can hurt the performance on the objective score; 3) a compromise by developing *soft* algorithms whereby the output of neural networks obeys a relaxed constraint, and there can still occur an arbitrary degree of constraint-violation. Towards the ultimate goal of establishing a general framework for neural CO solver with the ability to control an arbitrary-small degree of constraint violation, in this paper, we focus on a more achievable and common setting: the cardinality constraints, which in fact can be readily encoded by a differentiable optimal transport (OT) layer. Based on this observation, we propose OT-based cardinality constraint encoding for end-to-end CO problem learning with two variants: Sinkhorn and Gumbel-Sinkhorn, whereby their violation of the constraints can be exactly characterized and bounded by our theoretical results. On synthetic and real-world CO problem instances, our methods surpass the state-of-the-art CO network and are comparable to (if not superior to) the commercial solver Gurobi. In particular, we further showcase a case study of applying our approach to the predictive portfolio optimization task on real-world asset price data, improving the Sharpe ratio from 1.1 to 2.0 of a strong LSTM+Gurobi baseline under the classic predict-*then*-optimize paradigm.

## 1 Introduction

Developing neural networks that can handle combinatorial optimization (CO) problems is a trending research topic (Vinyals et al., 2015; Dai et al., 2016; Yu et al., 2020). A family of recent CO networks (Wang et al., 2019b; Li et al., 2019; Karalias & Loukas, 2020; Bai et al., 2019) improves the existing reinforcement learning-based auto-regressive CO networks (Dai et al., 2016; Lu et al., 2019) by solving the problem in one shot and relaxing the non-differentiable constraints, resulting in an end-to-end learning pipeline. The superiorities of one-shot CO networks are recognized in three aspects: 1) the higher efficiency by exploiting the GPU-friendly one-shot feed-forward network, compared to CPU-based traditional solvers (Gamrath et al., 2020) and the tedious auto-regressive

---

*Junchi Yan is the correspondence author. The work was in part supported by National Key Research and Development Program of China (2020AAA0107600), NSFC (U19B2035, 62222607, 61972250), STCSM (22511105100), Shanghai Committee of Science and Technology (21DZ1100100).

Table 1: Comparison among CO networks. Both theoretically and empirically, smaller constraint-violation (CV) leads to better optimization results. Logarithm terms in CV bounds are ignored.

| name of self-supervised CO network | Erdos Goes Neural (Karalias & Loukas, 2020) | CardNN-S (ours) | CardNN-GS/HGS (ours) |
|---|---|---|---|
| enforce constraint in network architecture | No (loss penalty term) | Yes (by Sinkhorn) | Yes (by Gumbel-Sinkhorn) |
| theoretical bound of CV (notations from Sec. 2) | non-controlled | $\widetilde{O}\left(\frac{m\tau}{\|\phi_k-\phi_{k+1}\|}\right)$ | $\widetilde{O}\left(\frac{m\tau(\|\phi_i-\phi_j\|+\sigma)}{\|\phi_i-\phi_j\|^2+\sigma^2}\right)_{\forall i\neq j}$ |
| empirical CV (results from Fig. 3(a)) | 8.44 | 6.71 | 0.09 |
| empirical optimal gap ($\downarrow$) | 0.152 | 0.139 | **0.023** |

CO networks; 2) the natural label-free, self-supervised learning paradigm by directly optimizing over the objective score, which is more practical than supervised learning (Vinyals et al., 2015) and empirically more efficient than reinforcement learning (Schulman et al., 2017); 3) the end-to-end architecture enabling tackling the important predictive CO problems, i.e. decision-making under uncertainty (Wilder et al., 2019; Elmachtoub & Grigas, 2022). In this paper, we follow the general paradigm of learning to solve CO in one-shot presented in the seminal work (Karalias & Loukas, 2020). A neural network CO solver is built upon a problem encoder network, which firstly accepts raw problem data and predicts the decision variables for the problem. The decision variables are then passed to a differentiable formula to estimate the objective score, and finally, the objective score is treated as the self-supervised loss. All modules must be differentiable for end-to-end learning.

As a CO solver, the output of the network should obey the constraint of the CO problem, while still preserving the gradient. Since the input-output mappings of CO are piece-wise constant, where the real gradient is zero almost everywhere or infinite when the output changes, it is notoriously hard to encode CO constraints in neural networks. There are three typical workarounds available: 1) In Karalias & Loukas (2020), the constraints are softly enforced by a penalty term, and the degree of constraint-violation can be hardly theoretically characterized nor controlled, which limits their applicability in many constraint-critical scenarios. Meanwhile, in the obligatory discretization step, adding penalty terms means that the algorithm must search a much larger space than if it was confined to feasible configurations, making the search less efficient and less generalizable (see Table 1). 2) The perturbation-based black-box differentiation methods (Pogančić et al., 2019; Paulus et al., 2021; Berthet et al., 2020) resorts to adding perturbation to the input-output mapping of discrete functions to estimate the approximate gradient as such the strict constraints are enforced in brute force, yet their approximate gradients may hurt the learning process. 3) The *soft* algorithms (Zanfir & Sminchisescu, 2018; Wang et al., 2019a; Sakaue, 2021) encode constraints to neural networks by developing approximate and differentiable algorithms for certain CO problems (graph matching, SAT, submodular), which is followed in this paper for their efficiency, yet there still remains the possibility of facing an arbitrary degree of constraint-violation.

Towards the ultimate goal of devising a general CO network solver addressing all the above issues, in this paper, we focus on developing a more practical paradigm for solving the cardinality-constrained CO problems (Buchbinder et al., 2014). The cardinality constraints $\|\mathbf{x}\|_0 \leq k$ are commonly found in a wide range of applications such as planning facility locations in business operation (Liu, 2009), discovering the most influential seed users in social networks (Chen et al., 2021), and predicting portfolios with controllable operational costs (Chang et al., 2000). Under the cardinality constraint, we aim to find the optimal subset with size $k$. Likewise other discrete CO constraints, the cardinality constraint is non-trivial to differentiate through. In this paper, we propose to encode cardinality constraints to CO networks by a top$k$ selection over a probability distribution (which is the output of an encoder network). An intuitive approach is to sort all probabilities and select the $k$-largest ones, however, such a process does not offer informative gradients. Inspired by Cuturi et al. (2019); Xie et al. (2020), we develop a *soft* algorithm by reformulating the top$k$ selection as an optimal transport problem (Villani, 2009) and efficiently tackle it by the differentiable Sinkhorn algorithm (Sinkhorn, 1964). With a follow-up differentiable computation of the self-supervised loss, we present a CO network whose output is *softly* cardinality-constrained and capable of end-to-end learning.

However, our theoretical characterization of the Sinkhorn-based *soft* algorithm shows its violation of the cardinality constraint may significantly grow if the values of the $k$-th and $(k+1)$-th probabilities are too close. Being aware of the perturbation-based differentiable methods (Pogančić et al., 2019; Paulus et al., 2021; Berthet et al., 2020) and the Gumbel trick (Jang et al., 2017; Mena et al., 2018; Grover et al., 2019) that can build near-discrete neural networks, in this paper, we further incorporate the Gumbel trick which is crucial for strictly bounding the constraint-violation term to an arbitrary

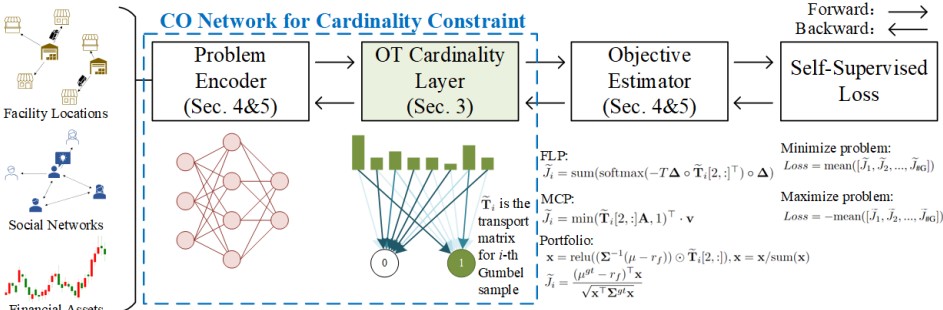

Figure 1: Our CardNN pipeline. The problem encoder and our proposed optimal transport (OT) cardinality layer compose our CO solver network, which has the superiority of guaranteeing a theoretically bounded constraint-violation. The decision variables from the CO network are then utilized to estimate the objective score, i.e. the self-supervised loss. The implementation of the OT cardinality layer and its theoretical characteristics will be discussed in Sec. 2. The other components are problem-dependent and will be discussed in Sec. 3 and Sec. 4 under the context of each problem.

small number. Our network takes both advantages of the high efficiency in *soft* algorithms (Zanfir & Sminchisescu, 2018) and the low constraint-violation in perturbation-based methods (Pogančić et al., 2019; Jang et al., 2017). A homotopy extension (Xu et al., 2016) is further developed where the constraint-violation term is gradually tightened. Following the self-supervised learning pipeline in Karalias & Loukas (2020), our cardinality-constrained CO networks are validated on two representative deterministic CO tasks: facility location and max covering problems.

An important application of predictive CO is also addressed, where the problem parameters are unknown at the decision-making time. We present a "predict-*and*-optimize" network that jointly learns a predictor and a neural network CO solver end-to-end over the final objective score, instead of the two-stage "predict-*then*-optimize" which learns a predictor first and then optimizes separately, at the risk of optimizing performance being hurt by prediction error. Specifically, towards a practical and widely concerned task: portfolio optimization under uncertainty, we build an end-to-end predictive portfolio optimization model. Experimental results on real-world data show that it outperforms the classic "predict-*then*-optimize" paradigm. **The contributions include**:

• **New End-to-end One-shot Neural Architecture for CO Problems.** We propose the first (to our best knowledge) end-to-end cardinality-constrained neural network for efficient CO problem-solving in one-shot, in the sense that the constraints are incorporated in the network architecture instead of directly putting them in the learning objective as penalty terms.

• **Theoretical and Empirical Advantages of the CO Architecture.** The cardinality constraint is encoded in the differentiable optimal transport layer based on the top$k$ selection technique (Xie et al., 2020). While we further introduce the idea of perturbation as used in blackbox differentiable CO (Pogančić et al., 2019; Paulus et al., 2021), by incorporating the Gumbel trick to reduce the constraint-violation, and the violation bound is strictly guaranteed by our theoretical results. Empirical results on two CO tasks: facility location and max covering also verify its competitiveness.

• **Enabling "predict-*and*-optimize" Paradigm.** We show that our new network further enables an emerging end-to-end "predict-*and*-optimize" paradigm in contrast to the traditional "predict-*then*-optimize" pipeline. Its potential is demonstrated by a study on predictive portfolio optimization on real-world asset price data, with an improvement of Sharpe ratio from 1.1 to 2.0, compared with a baseline: LSTM+Gurobi.

## 2 CARDINLIATY-CONSTRAINED COMBINATORIAL NETWORKS

An overview of our CardNN pipeline is shown in Fig. 1. Following the general paradigm (Karalias & Loukas, 2020) to tackle CO in one-shot, we introduce an optimal transport (OT) cardinality layer in the neural network CO solver to enforce the constraints upon the output of the problem encoder network, whereby the superiorities could be addressed both empirically and theoretically.

Recall that under cardinality constraint, the solution must have no more than $k$ non-zero elements:

$$\min_{\mathbf{x}} J(\mathbf{x}) \qquad s.t. \quad \|\mathbf{x}\|_0 \le k. \tag{1}$$

In this paper, enforcing the cardinality constraint in networks is formulated as solving OT with differentiable layers (Cuturi, 2013). Denote $\mathbf{s} = [s_1, \cdots, s_m]$ as the probability vector predicted by the problem encoder network, our OT layer selects $k$ largest items from $\mathbf{s}$ by moving $k$ items to one destination (selected), and the other $(m - k)$ elements to the other destination (not selected). In the following, we present two embodiments of OT layers and their theoretical characteristics.

## 2.1 CardNN-S: Sinkhorn Layer for Cardinality Constraint

We follow the popular method Sinkhorn (1964) and define the OT problem as follows. The sources are $m$ candidates in $\mathbf{s}$ and the destinations are the min/max values of $\mathbf{s}$. OT moves the top-$k$ items to $s_{\text{max}}$, and the others to $s_{\text{min}}$. The marginal distributions $(\mathbf{c}, \mathbf{r})$ and distance matrix $(\mathbf{D})$ are defined as:

$$\mathbf{c} = \underbrace{[1 \quad 1 \quad ... \quad 1]}_{m \text{ items}}, \mathbf{r} = \begin{bmatrix} m - k \\ k \end{bmatrix}, \mathbf{D} = \begin{bmatrix} s_1 - s_{\text{min}} & s_2 - s_{\text{min}} & ... & s_m - s_{\text{min}} \\ s_{\text{max}} - s_1 & s_{\text{max}} - s_2 & ... & s_{\text{max}} - s_m \end{bmatrix}. \quad (2)$$

Then OT can be formulated as integer linear programming:

$$\min_{\mathbf{T}} \operatorname{tr}(\mathbf{T}^\top \mathbf{D}) \qquad s.t. \quad \mathbf{T} \in \{0, 1\}^{2 \times m}, \mathbf{T}\mathbf{1} = \mathbf{r}, \mathbf{T}^\top \mathbf{1} = \mathbf{c}, \quad (3)$$

where $\mathbf{T}$ is the transportation matrix which is also a feasible decision variable for the cardinality constraint, and $\mathbf{1}$ is a column vector whose all elements are 1s. The optimal solution $\mathbf{T}^*$ to Eq. (3) should be equivalent to the solution by firstly sorting all items and then selecting the top-$k$ items. To make the process differentiable by *soft* algorithms, the binary constraint on $\mathbf{T}$ is relaxed to continuous values $[0, 1]$, and Eq. (3) is modified with an entropic regularization:

$$\min_{\mathbf{T}^\tau} \operatorname{tr}(\mathbf{T}^{\tau\top} \mathbf{D}) + \tau h(\mathbf{T}^\tau) \qquad s.t. \quad \mathbf{T}^\tau \in [0, 1]^{2 \times m}, \mathbf{T}^\tau \mathbf{1} = \mathbf{r}, \mathbf{T}^{\tau\top} \mathbf{1} = \mathbf{c}, \quad (4)$$

where $h(\mathbf{T}^\tau) = \sum_{i,j} \mathbf{T}_{ij}^\tau \log \mathbf{T}_{ij}^\tau$ is the entropic regularizer (Cuturi, 2013). Given any real-valued matrix $\mathbf{D}$, Eq. (4) is solved by firstly enforcing the regularization factor $\tau$: $\mathbf{T}^\tau = \exp(-\mathbf{D}/\tau)$. Then $\mathbf{T}^\tau$ is row- and column-wise normalized alternatively:

$$\mathbf{D}_r = \operatorname{diag}(\mathbf{T}^\tau \mathbf{1} \oslash \mathbf{r}), \ \mathbf{T}^\tau = \mathbf{D}_r^{-1} \mathbf{T}^\tau; \quad \mathbf{D}_c = \operatorname{diag}(\mathbf{T}^{\tau\top} \mathbf{1} \oslash \mathbf{c}), \ \mathbf{T}^\tau = \mathbf{T}^\tau \mathbf{D}_c^{-1}, \quad (5)$$

where $\oslash$ is element-wise division. We denote $\mathbf{T}^{\tau*}$ as the converged solution, which is the optimal solution to Eq. (4). The second row of $\mathbf{T}^{\tau*}$ is regarded as the relaxed decision variable for the cardinality constraint: $\mathbf{T}^{\tau*}[2, i]$ is regarded as the probability that $x_i$ should be non-zero. $\mathbf{T}^{\tau*}$ is further fed into the objective estimator. Note that $\mathbf{T}^{\tau*}$ is usually infeasible in the original problem, and we define the following constraint violation to measure the quality of $\mathbf{T}^{\tau*}$.

**Definition 2.1** (Constraint Violation, CV). CV is the expected minimal distance between a relaxed solution $\mathbf{t}$ (from distribution $\mathcal{T}$) and any feasible solution from the feasible set $\mathcal{H}$: $CV = \mathbb{E}_{\mathbf{t} \in \mathcal{T}} [\min_{\mathbf{h} \in \mathcal{H}} \|\mathbf{t} - \mathbf{h}\|_F]$. Apparently, $\mathbf{h}$ is the nearest feasible solution to $\mathbf{t}$.

*Remark* 2.2 (Meaning of CV). Take the self-supervised CardNN-S as an example, estimating the objective score (which is exactly the self-supervised loss) based on $\mathbf{T}^{\tau*}$ is necessary during training. In inference, the solution must be feasible in the original problem, so the nearest feasible solution $\mathbf{T}^*$ is returned. Actually, in training, the network learns to solve a relaxed, easier version of the original problem, and $CV = \|\mathbf{T}^* - \mathbf{T}^{\tau*}\|_F$ is an important characteristic measuring the gap between the relaxed problem (in training) and the original problem (in inference). Here $\mathcal{T}$ means the distribution of all CardNN-S outputs and is omitted for simplicity. Such a meaning of CV also applies for other self-supervised CO networks. In the following, we theoretically characterize the CV of CardNN-S:

**Proposition 2.3.** *Assume that Sinkhorn is converged. The constraint-violation of the CardNN-S is*

$$CV_{CardNN\text{-}S} = \|\mathbf{T}^* - \mathbf{T}^{\tau*}\|_F \leq \frac{2m\tau \log 2}{|\phi_k - \phi_{k+1}|}. \quad (6)$$

Without loss of generality, $\phi$ is denoted as the descending sequence of $\mathbf{s}$, i.e. $\phi_k, \phi_{k+1}$ are the $k$-th, $(k + 1)$-th largest elements of $\mathbf{s}$, respectively. Proposition 2.3 is a straightforward derivation based on Theorem 2 of Xie et al. (2020), and is better than Karalias & Loukas (2020) whose CV is non-controlled. However, as we learn from Eq. (6), the CV of CardNN-S gradually grows if $|\phi_k - \phi_{k+1}|$ becomes smaller, and turns diverged under the extreme case that $\phi_k = \phi_{k+1}$, meaning that its CV cannot be tighten by adjusting the hyperparameter $\tau$. Such a divergence is not surprising

---

**Algorithm 1: CardNN-GS: Gumbel-Sinkhorn Layer for Cardinality Constraint**

**Input:** List $\mathbf{s}$ with $m$ items; cardinality $k$; Sinkhorn factor $\tau$; noise factor $\sigma$; sample size #G.

1 **for** $i \in \{1, 2, ..., \#G\}$ **do**

2      for all $s_j$, $\widetilde{s}_j = s_j - \sigma \log(-\log(u_j))$, where $u_j$ is from $(0, 1)$ uniform distribution;

3      $\widetilde{\mathbf{D}} = \begin{bmatrix} \widetilde{s}_1 - s_{\mathtt{min}} & ... & \widetilde{s}_m - s_{\mathtt{min}} \\ s_{\mathtt{max}} - \widetilde{s}_1 & ... & s_{\mathtt{max}} - \widetilde{s}_m \end{bmatrix}$; construct $\mathbf{c}, \mathbf{r}$ following Eq. (2); $\widetilde{\mathbf{T}}_i = \exp(-\widetilde{\mathbf{D}}/\tau)$;

4      **while** *not converged* **do**

5          $\widetilde{\mathbf{D}}_r = \mathrm{diag}(\widetilde{\mathbf{T}}_i \mathbf{1} \oslash \mathbf{r})$; $\widetilde{\mathbf{T}}_i = \widetilde{\mathbf{D}}_r^{-1} \widetilde{\mathbf{T}}_i$; $\widetilde{\mathbf{D}}_c = \mathrm{diag}(\widetilde{\mathbf{T}}_i^\top \mathbf{1} \oslash \mathbf{c})$; $\widetilde{\mathbf{T}}_i = \widetilde{\mathbf{T}}_i \widetilde{\mathbf{D}}_c^{-1}$;

**Output:** A list of transport matrices $[\widetilde{\mathbf{T}}_1, \widetilde{\mathbf{T}}_2, ..., \widetilde{\mathbf{T}}_{\#G}]$.

---

because one cannot decide whether to select $\phi_k$ or $\phi_{k+1}$ if they are equal, which is fine if any direct supervision on $\mathbf{T}^{\tau*}$ is available. However, as discussed in Remark 2.2, the importance of CV is non-negligible in self-supervised CO networks. Since working with solely the Sinkhorn algorithm reaches its theoretical bottleneck, in the following, we present our improved version by introducing random perturbations (Pogančić et al., 2019; Jang et al., 2017) to further tighten the CV.

## 2.2 CARDNN-GS: GUMBEL-SINKHORN LAYER FOR CARDINALITY CONSTRAINT

In this section, we present our Gumbel-Sinkhorn Layer for Cardinality Constraint as summarized in Alg. 1 and we will theoretically characterize its CV. Following the reparameterization trick (Jang et al., 2017), instead of sampling from a distribution that is non-differentiable, we add random variables to probabilities predicted by neural networks. The Gumbel distribution is:

$$g_\sigma(u) = -\sigma \log(-\log(u)), \tag{7}$$

where $\sigma$ controls the variance and $u$ is from $(0, 1)$ uniform distribution. We can update $\mathbf{s}$ and $\mathbf{D}$ as:

$$\widetilde{s}_j = s_j + g_\sigma(u_j), \quad \widetilde{\mathbf{D}} = \begin{bmatrix} \widetilde{s}_1 - s_{\mathtt{min}} & \widetilde{s}_2 - s_{\mathtt{min}} & ... & \widetilde{s}_m - s_{\mathtt{min}} \\ s_{\mathtt{max}} - \widetilde{s}_1 & s_{\mathtt{max}} - \widetilde{s}_2 & ... & s_{\mathtt{max}} - \widetilde{s}_m \end{bmatrix}. \tag{8}$$

Again we formulate the integer linear programming version of the OT with Gumbel noise:

$$\min_{\mathbf{T}^\sigma} \mathrm{tr}(\mathbf{T}^{\sigma\top} \widetilde{\mathbf{D}}) \qquad s.t. \quad \mathbf{T}^\sigma \in \{0, 1\}^{2 \times m}, \mathbf{T}^\sigma \mathbf{1} = \mathbf{r}, \mathbf{T}^{\sigma\top} \mathbf{1} = \mathbf{c}, \tag{9}$$

where the optimal solution to Eq. (9) is denoted as $\mathbf{T}^{\sigma*}$. To make the integer linear programming problem feasible for gradient-based deep learning methods, we also relax the integer constraint and add the entropic regularization term:

$$\min_{\widetilde{\mathbf{T}}} \mathrm{tr}(\widetilde{\mathbf{T}}^\top \widetilde{\mathbf{D}}) + h(\widetilde{\mathbf{T}}) \qquad s.t. \quad \widetilde{\mathbf{T}} \in [0, 1]^{2 \times m}, \widetilde{\mathbf{T}} \mathbf{1} = \mathbf{r}, \widetilde{\mathbf{T}}^\top \mathbf{1} = \mathbf{c}, \tag{10}$$

which is tackled by the Sinkhorn algorithm following Eq. (5). Here we denote the optimal solution to Eq. (10) as $\widetilde{\mathbf{T}}^*$. Since $\mathbf{T}^{\sigma*}$ is the nearest feasible solution to $\widetilde{\mathbf{T}}^*$, we characterize the constraint-violation as the expectation of $\|\mathbf{T}^{\sigma*} - \widetilde{\mathbf{T}}^*\|_F$, and multiple $\widetilde{\mathbf{T}}$s are generated in parallel in practice to overcome the randomness (note that $\phi$ is the descending ordered version of $\mathbf{s}$):

**Proposition 2.4.** *With probability at least $(1 - \epsilon)$, the constraint-violation of the CardNN-GS is*

$$CV_{\text{CardNN-GS}} = \mathbb{E}_u \left[ \|\mathbf{T}^{\sigma*} - \widetilde{\mathbf{T}}^*\|_F \right] \le (\log 2) m \tau \sum_{i \ne j} \Omega(\phi_i, \phi_j, \sigma, \epsilon), \tag{11}$$

$$\text{where } \Omega(\phi_i, \phi_j, \sigma, \epsilon) = \frac{2\sigma \log\left(\sigma - \frac{|\phi_i - \phi_j| + 2\sigma}{\log(1 - \epsilon)}\right) + |\phi_i - \phi_j| \left(\frac{\pi}{2} + \arctan \frac{\phi_i - \phi_j}{2\sigma}\right)}{(1 - \epsilon)((\phi_i - \phi_j)^2 + 4\sigma^2)(1 + \exp \frac{\phi_i - \phi_k}{\sigma})(1 + \exp \frac{\phi_{k+1} - \phi_j}{\sigma})}.$$

*Proof sketch:* This proposition is proven by generalizing Proposition 2.3. We denote $\phi_{\pi_k}, \phi_{\pi_{k+1}}$ as the $k$-th and $(k + 1)$-th largest items after perturbed by the Gumbel noise, and our aim becomes to prove the upper bound of $\mathbb{E}_u \left[ 1/(|\phi_{\pi_k} + g_\sigma(u_{\pi_k}) - \phi_{\pi_{k+1}} - g_\sigma(u_{\pi_{k+1}})|) \right]$, where the probability density function of $g_\sigma(u_{\pi_k}) - g_\sigma(u_{\pi_{k+1}})$ can be bounded by $f(y) = 1/(y^2 + 4)$. Thus we can compute the bound by integration. See Appendix C.1 for details. □

Table 2: Objective score↑ among perturb-based methods (Pogančić et al., 2019; Berthet et al., 2020; Amos et al., 2019) on MCP (k=50,m=500,n=1000). Baseline is Xie et al. (2020) used in CardNN-S.

| CardNN+Pogančić et al. (2019) | CardNN+Berthet et al. (2020) | CardNN+Amos et al. (2019) | CardNN-S | CardNN-GS |
|---|---|---|---|---|
| 32499.7 | 37618.9 | 38899.6 | 42034.9 | 44710.3 |

**Corollary 2.5.** *Ignoring logarithm terms for simplicity,* $CV_{CardNN\text{-}GS} \leq \widetilde{O}\left(\frac{m\tau(|\phi_i-\phi_j|+\sigma)}{|\phi_i-\phi_j|^2+\sigma^2}\right)_{\forall i \neq j}$ *(see Appendix C.2 for the proof).*

We compare CV of CardNN-S and CardNN-GS by the toy example in Fig. 2: finding the top3 of $[1.0, 0.8, 0.601, 0.6, 0.4, 0.2]$. We plot CV w.r.t. different $\tau, \sigma$ values. CV is tightened by larger $\sigma$ and smaller $\tau$ for CardNN-GS, compared to CardNN-S whose violation is larger and can only be controlled by $\tau$. These empirical results are in line with Proposition 2.3 and Proposition 2.4.

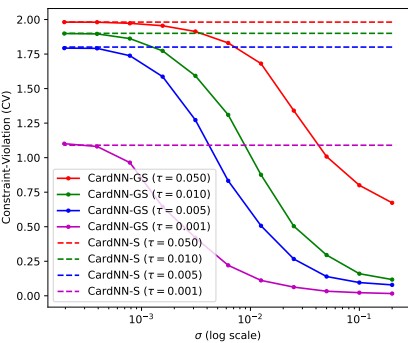

Figure 2: Toy example.

**Homotopy Gumbel-Sinkhorn**. The Corollary 2.5 suggests that CV can be tightened by adjusting $\tau$ and $\sigma$, motivating us to develop a homotopy (Xiao & Zhang, 2013; Xu et al., 2016) Gumbel-Sinkhorn method where the constraints are gradually tighten (i.e. annealing $\tau$ and $\sigma$ values). In practice, $\sigma$ is not considered because a larger $\sigma$ means increased variance which calls for more Gumbel samples. We name the homotopy version as CardNN-HGS.

We also notice that our CardNN-S (Sec. 2.1) and CardNN-GS (Sec. 2.2) can be unified theoretically:

**Corollary 2.6.** *CardNN-S is a special case of CardNN-GS when $\sigma \to 0^+$ (proof in Appendix C.3).*

## 3 ONE-SHOT SOLVING THE DETERMINISTIC CO TASKS

In this section, we present the implementation details and experiment results for learning to solve two deterministic CO problems in one-shot: facility location problem (FLP) and max covering problem (MCP). Deterministic CO means all problem parameters are known at the decision-making time. Readers are referred to Appendix D for the algorithm details.

**The Facility Location Problem**. Given $m$ locations and we want to extend $k$ facilities such that the goods can be stored at the nearest facility and delivered more efficiently (Liu, 2009). The objective is to minimize the sum of the distances between each location and its nearest facility.
*Problem Formulation:* Denote $\mathbf{\Delta} \in \mathbb{R}_{\geq 0}^{m \times m}$ as the distance matrix for locations, the FLP is

$$\min_{\mathbf{x}} \sum_{j=1}^{m} \min(\{\mathbf{\Delta}_{i,j} | \forall \mathbf{x}_i = 1\}) \qquad s.t. \quad \mathbf{x} \in \{0,1\}^m, \|\mathbf{x}\|_0 \leq k. \tag{12}$$

*Problem Encoder:* For locations with 2-D coordinates, an edge is defined if two locations are closer than a threshold, e.g. 0.02. We exploit a 3-layer SplineCNN (Fey et al., 2018) to extract features.
*Objective Estimator:* We notice that the $\min$ operator in Eq. (12) will lead to sparse gradients. Denote $\circ$ as element-wise product of a matrix and a tiled vector, we replace $\min$ by Softmax with minus temperature $-\beta$: $\widetilde{J}_i = \text{sum}(\text{softmax}(-\beta\mathbf{\Delta} \circ \widetilde{\mathbf{T}}_i[2,:]^\top) \circ \mathbf{\Delta})$, $J = \text{mean}([\widetilde{J}_1, \widetilde{J}_2, ..., \widetilde{J}_{\#G}])$.

**The Max Covering Problem**. Given $m$ sets and $n$ objects where each set may cover any number of objects, and each object is associated with a value, MCP (Khuller et al., 1999) aims to find $k$ sets ($k \ll m$) such that the covered objects have the maximum sum of values. This problem reflects real-world scenarios such as discovering influential seed users in social networks (Chen et al., 2021).
*Problem Formulation:* We build a bipartite graph for the sets and objects, whereby coverings are encoded as edges. Denote $\mathbf{v} \in \mathbb{R}^n$ as the values, $\mathbf{A} \in \{0,1\}^{m \times n}$ as the adjacency of bipartite graph, $\mathbb{I}(\mathbf{x})$ as an indicator $\mathbb{I}(\mathbf{x})_i = 1$ if $\mathbf{x}_i \geq 1$ else $\mathbb{I}(\mathbf{x})_i = 0$. We formulate the MCP as

$$\max_{\mathbf{x}} \sum_{j=1}^{n} \left( \mathbb{I}\left(\sum_{i=1}^{m} \mathbf{x}_i \mathbf{A}_{ij}\right) \cdot \mathbf{v}_j \right) \qquad s.t. \quad \mathbf{x} \in \{0,1\}^m, \|\mathbf{x}\|_0 \leq k, \tag{13}$$

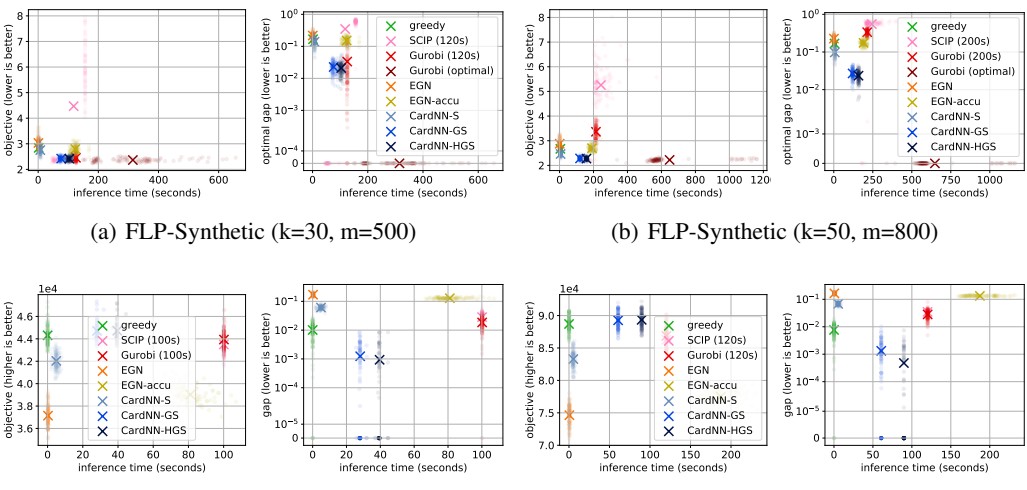

(a) FLP-Synthetic (k=30, m=500)  (b) FLP-Synthetic (k=50, m=800)

(c) MCP-Synthetic (k=50, m=500, n=1000)  (d) MCP-Synthetic (k=100, m=1000, n=2000)

Figure 3: Plot of objective score, gap w.r.t. inference time on synthetic CO problems. Each scatter dot denotes a problem instance, and the average performance is marked by "×". In terms of both efficiency and efficacy, our CardNN-S outperforms the EGN CO network whose constraint-violation is non-controlled. The efficacy is further improved by CardNN-GS and CardNN-HGS, even surpassing the state-of-the-art commercial solver Gurobi (better results with less inference time). The Gurobi solver fails to return the optimal solution within 24 hours for MCP, thus not reported here.

*Problem Encoder:* To encode the bipartite graph, we exploit three layers of GraphSage (Hamilton et al., 2017) followed by a fully-connected layer with sigmoid to predict the probability of selecting each set.

*Objective Estimator:* Based on Eq. (13), the objective value is estimated as:

$$\widetilde{J}_i = \min(\widetilde{\mathbf{T}}_i[2,:]\mathbf{A}, 1)^\top \cdot \mathbf{v}, \ J = \mathrm{mean}([\widetilde{J}_1, \widetilde{J}_2, ..., \widetilde{J}_{\#\mathrm{G}}]). \tag{14}$$

**Learning and Optimization**. Based on whether it is a minimization or a maximization problem, $J$ or $-J$ is treated as the self-supervised loss, respectively. The Adam optimizer (Kingma & Ba, 2014) is applied for training. In inference, the neural network prediction is regarded as initialization, and we also optimize the probabilities w.r.t. the objective score by gradients.

**Experiment Setup**. We follow the self-supervised learning pipeline proposed by the state-of-the-art CO network (Karalias & Loukas, 2020), whereby both synthetic data and real-world data are considered. For synthetic data, we build separate training/testing datasets with 100 samples. We generate uniform random locations on a unit square for FLP, and we follow the distribution in OR-LIB (Beasley, 1990) for MCP. Due to the lack of large-scale datasets, real-world datasets are only considered for testing (training on synthetic data, testing on real-world data). We test the FLP based on Starbucks locations in 4 cities worldwide with 166-569 stores, and we test MCP based on 6 social networks with 1912-9498 nodes collected from Twitch by Rozemberczki et al. (2021).

**Baselines**. **1) Greedy algorithms** are considered because they are easy to implement but very strong and effective. They have the worst-case approximation ratio of $(1 - 1/e)$ due to the submodular property (Fujishige, 1991) for both FLP and MCP. **2) Integer programming solvers** including the state-of-the-art commercial solver Gurobi 9.0 (Gurobi Optimization, LLC, 2021) and the state-of-the-art open-source solver SCIP 7.0 (Gamrath et al., 2020). The time budgets of solvers are set to be higher than our networks. For **3) CO neural networks**, we compare with the state-of-the-art Erdos Goes Neural (EGN) (Karalias & Loukas, 2020) which is adapted from their official implementation: `https://github.com/Stalence/erdos_neu`. The major difference between EGN and ours is that EGN does not enforce CO constraints by its architecture. Besides, we empirically find out that all self-supervised learning methods converge within tens of minutes. Since the RL methods e.g. Khalil et al. (2017); Wang et al. (2021a) need much more training time, they are not compared.

**Metrics and Results**. Fig. 3 and 4 report results on synthetic and real-world dataset, respectively. The "gap" metric is computed as gap $= |J - J^*|/\max(J, J^*)$, where $J$ is the predicted objective

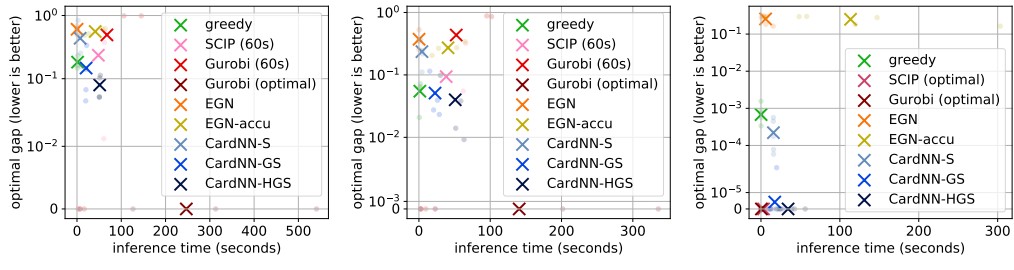

(a) FLP-Starbucks (Euclidean)  (b) FLP-Starbucks (Manhattan)  (c) MCP-Twitch

Figure 4: Plot of optimal gap w.r.t. inference time on real-world CO problems. Our CardNN models are consistently superior than EGN, and are comparative to state-of-the-art SCIP/Gurobi solvers and sometimes can even surpass. On the FLP-Starbucks problems, our CardNN-GS/HGS achieve a lower optimal gap with comparable time cost w.r.t. SCIP/Gurobi. On the MCP-Twitch problems, our CardNN-HGS is slower than SCIP/Gurobi, but it finds all optimal solutions.

and $J^*$ is the incumbent best objective value (among all methods). If one of the integer programming solvers proves an optimal solution, we name it as "optimal gap". Considering both efficiency and efficacy, the performance ranking of CO networks is CardNN-HGS > CardNN-GS > CardNN-S > EGN. This is in line with our theoretical result in Sec. 2: a lower constraint violation will lead to better performance in CO. To justify our selection of Xie et al. (2020) as the base differentiable method, we also implement other perturbation-based differentiable methods and report the MCP results in Table 2. See Appendix E for more details about our deterministic CO experiment.

## 4 ONE-SHOT SOLVING THE PREDICTIVE CO TASKS

In this section, we study the interesting and important topic of predictive CO problems where the problem parameters are unknown at the decision-making time. We consider the challenging problem of predicting the portfolio with the best trade-off in risks and returns in the future, under the practical cardinality constraint to control the operational costs. Traditionally, such a problem involves two separate steps: 1) predict the asset prices in the future, probably by some deep learning models; 2) find the best portfolio by solving an optimization problem based on the prediction. However, the optimization process may be misled due to unavoidable errors in the prediction model. To resolve this issue, Solin et al. (2019) proposes to differentiate through unconstrained portfolio optimization via Amos & Kolter (2017), but the more practical cardinality constrained problem is less studied.

**Problem Formulation**. Cardinality constrained portfolio optimization considers a practical scenario where a portfolio must contain no more than $k$ assets (Chang et al., 2000). A good portfolio aims to have a high return (measured by mean vector $\mu \in \mathbb{R}^m$) and low risk (covariance matrix $\boldsymbol{\Sigma} \in \mathbb{R}^{m \times m}$). Here we refer to maximizing the Sharpe ratio (Sharpe, 1998). The problem is formulated as

$$\max_{\mathbf{x}} \frac{(\mu - r_f)^\top \mathbf{x}}{\sqrt{\mathbf{x}^\top \boldsymbol{\Sigma} \mathbf{x}}}, \qquad s.t. \quad \sum_{i=1}^m x_i = 1, \mathbf{x} \geq 0, \|\mathbf{x}\|_0 \leq k, \tag{15}$$

where $\mathbf{x}$ denotes the weight of each asset, $r_f$ means risk-free return, e.g. U.S. treasury bonds. Note that $\mu, \boldsymbol{\Sigma}$ are unknown at the time of decision-making, and they are predicted by a neural network.

**Network Architecture**. An encoder-decoder architecture of Long-Short Term Memory (LSTM) modules is adopted as the problem encoder (i.e. price predictor). The sequence of historical daily prices is fed into the encoder module, and the decoder module outputs the predicted prices for the future. We append a fully-connected layer after the hidden states to learn the probabilities for cardinality constraints, followed by our CardNN-GS layers.

**Objective Estimator**. Based on the network outputs $\mu, \boldsymbol{\Sigma}, \widetilde{\mathbf{T}}$, we estimate the value of $\mathbf{x}$ by leveraging a closed-form solution of unconstrained Eq. (15): $\mathbf{x} = \boldsymbol{\Sigma}^{-1}(\mu - r_f)$, and then enforcing the constraints: $\mathbf{x} = \mathrm{relu}(\mathbf{x} \odot \widetilde{\mathbf{T}}_i[2,:]), \mathbf{x} = \mathbf{x}/\mathrm{sum}(\mathbf{x})$ ($\odot$ means element-wise product). After obtaining $\mathbf{x}$, we compute the Sharpe ratio based on $\mathbf{x}$ and $\mu^{gt}, \boldsymbol{\Sigma}^{gt}$ computed from the ground truth prices, and use this Sharpe ratio as supervision: $\widetilde{J}_i = \left((\mu^{gt} - r_f)^\top \mathbf{x}\right) / \left(\sqrt{\mathbf{x}^\top \boldsymbol{\Sigma}^{gt} \mathbf{x}}\right)$.

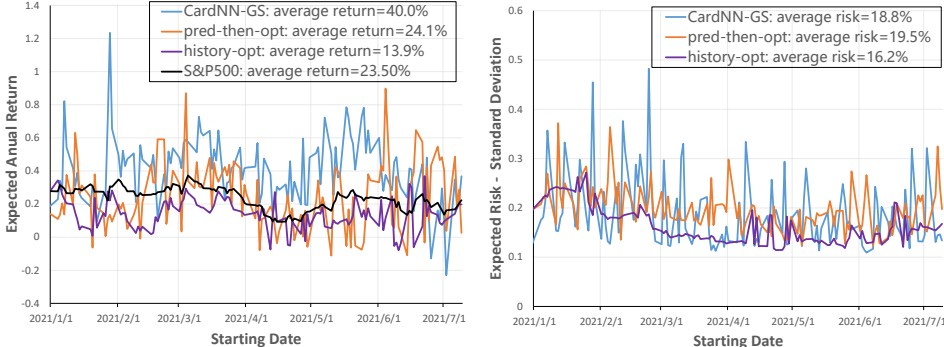

Figure 5: Return (left) and risk (right) for portfolios by the classic "predict-*then*-optimize" pipeline (by LSTM for prediction and Gurobi for optimization) and our CardNN-GS for end-to-end "predict-*and*-optimize". The portfolios proposed by our CardNN-GS has higher returns and lower risks. Since the batch of S&P500 assets violates the cardinality constraint, it is unfair to compare the risk.

Table 3: Our "predict-*and*-optimize" achieves better risk-return trade-off (Sharpe ratio) though the price prediction is less accurate (by mean square error) than "predict-*then*-optimize" on test set.

| Methods | predictor+optimizer | prediction MSE ↓ | Sharpe ratio ↑ |
|---|---|---|---|
| history-opt | none+Gurobi | (no prediction) | 0.673 |
| pred-*then*-opt | LSTM+Gurobi | **0.153** | 1.082 |
| pred-*and*-opt | LSTM+CardNN-GS | 1.382 | **1.968** |

**Setup and Baselines**. The price predictor is supervised with price labels but the optimizer is self-supervised (no optimal solution labels). We consider portfolio prediction with the best Sharpe ratio in the next 120 trading days (∼24 weeks) and test with the real data in the year 2021. The training set is built based on the prices of 494 assets from the S&P 500 index from 2018-01-01 to 2020-12-30. We set the annual risk-free return as 3% and the cardinality constraint $k = 20$. The classic "predict-*then*-optimize" baseline learns the same LSTM model as ours to minimize the prediction square error of the asset prices and optimizes the portfolio by Gurobi based on the price predictions. We also consider a "history-opt" baseline, whereby the optimal portfolio in historical data is followed.

**Results.** The portfolios are tested on the real data from 01-01-2021 to 12-30-2021, and the results are listed in Fig. 5 and Table 3. On average, we improve the annual return of the portfolio from 24.1% to 40%. The MSE in Table 3 denotes the mean square error of price predictions, and note that more accurate price predictions do not lead to better portfolios. We visualize the predicted portfolios in Fig. 6 and compare it to the efficient frontier (portfolios with optimal risk-return trade-off). Being closer to the frontier means a better portfolio. Also, note that reaching the efficient frontier is nearly impossible as the prediction always contains errors.

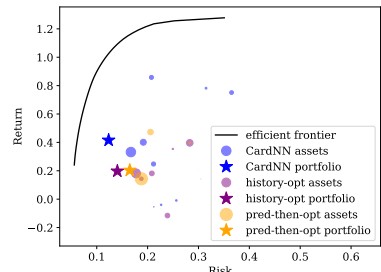

Figure 6: Visualization on 2021-03-25 data of individual assets. Larger dots mean higher weights. See more visualizations in Appendix G.

## 5 CONCLUSIONS

Towards the ultimate goal of developing general paradigms to encode CO constraints into neural networks with controlled constraint-violation bounds, in this paper, we have presented a differentiable neural network for cardinality-constrained combinatorial optimization. We theoretically characterize the constraint-violation of the Sinkhorn network (Sec. 2.1), and we introduce the Gumbel trick to mitigate the constraint-violation issue (Sec. 2.2). Our method is validated in learning to solve deterministic CO problems (on both synthetic and real-world problems) and end-to-end learning of predictive CO problems under the important predict-*and*-optimize paradigm.

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

## A   RELATED WORK

**CO Networks with Constraints Handling for Deterministic CO**. Multi-step methods encode constraints by manually programmed action spaces, and the networks can be learned by supervised labels (Vinyals et al., 2015) or by reinforcement learning (Khalil et al., 2017; Zhang et al., 2020; Chen & Tian, 2019). Controlling constraint-violation is less an issue for supervised or reinforcement learning because the supervision signals are directly passed to the output of neural networks. One-shot CO networks construct the solution by a single forward pass thus being more efficient. The seminal work (Karalias & Loukas, 2020) aims to develop a general pipeline for one-shot CO networks, by softly absorbing the violation of constraint as part of its final loss. However, our analysis shows that such a non-controlled constraint-violation potentially harms problem-solving. There also exist embodiments of constrained CO networks for tailored problems, e.g. the constraint can be encoded as doubly-stochastic matrices in assignment problems (Nowak et al., 2018; Wang et al., 2021b; 2020). These methods can be viewed as special cases of our paradigm (yet the powerful perturbation method is not fully exploited). Liu et al. (2019) can be viewed as a multi-step optimization variant of ours, yet learning is not considered and the constraint-violation issue is not theoretically characterized.

**Differentiable Optimizers for Predictive CO**. The major challenge of joint prediction and optimization is making the optimizers differentiable. In Amos & Kolter (2017); Agrawal et al. (2019), a family of convex optimization problems is found feasible to differentiate by their KKT conditions. However, for non-convex CO problems, Pogančić et al. (2019) explains that the true gradients are meaningless for neural networks. One family of papers propose to incorporate existing solvers and estimate the *informative and approximate* gradients, including tailoring *soft* algorithms for specific problems (Zanfir & Sminchisescu, 2018; Wang et al., 2019a; Sakaue, 2021) , or following the perturbation-based blackbox optimization pipeline (Pogančić et al., 2019; Paulus et al., 2021; Berthet et al., 2020) with certain restrictions such as the formula must be linear. The other paradigm is incorporating neural-network solvers which are naturally differentiable. For example, for graph matching on images (Fey et al., 2020; Sarlin et al., 2020), deep feature predictors and neural matching solvers are learned end-to-end under supervised learning, and their neural solvers leverage the Sinkhorn algorithm (Cuturi, 2013) as a neural network layer. In this paper, our neural network solver is incorporated for the new predictive portfolio optimization task, and our predict-*and*-optimize pipeline does not require the ground truth labels for the optimization problem.

## B   LIMITATIONS

We are also aware of the following **limitations**:

1) Our theoretical analysis mainly focuses on characterizing the constraint-violation. There is an unexplored theoretical aspect about the approximation ratios of our CO networks w.r.t. the optimal solution and the optimal objective score, and we plan to study it in future work.

2) The original EGN pipeline is relatively general for all constraints, and we restrict the scope of this paper within cardinality constraints. We are aware of a potential direction to extend our paper: the cardinality constraints are handled by our method (encoded in the network's output), and the other constraints are handled in a way similar to EGN (encoded as Lagrange multipliers or penalty terms). In such a sense, the cardinality constraints are handled efficiently while still preserving the generality of EGN.

3) In the predictive CO tasks, the predictor may be, in some degree, coupled with the follow-up neural network solver. In our predictive portfolio optimization experiment, our price predictor cannot generalize soundly for the Gurobi solver, and the Sharpe ratio degenerates to 1.002 if our price predictions are passed to Gurobi.

## C   PROOF OF THEOREMS

Before starting the detailed proof of the propositions and corollaries, firstly we recall the notations used in this paper:

- $\mathbf{T}^* = \mathrm{TopK}(\mathbf{D})$ is the optimal solution of the integer linear programming form of the OT problem Eq. (3), which is equivalent to the solution by firstly sorting all items and then selecting the top$k$ items. If the $k$-th and $(k+1)$-th largest items are equal, the algorithm randomly selects one to strictly satisfy the cardinality constraint;

- $\mathbf{T}^{\tau*} = \mathrm{Sinkhorn}(\mathbf{D})$ is the optimal solution of the entropic regularized form of the OT problem Eq. (4) solved by Sinkhorn algorithm. It is also the output by CardNN-S;

- $\mathbf{T}^{\sigma*} = \mathrm{TopK}(\widetilde{\mathbf{D}})$ is the optimal solution to the integer linear programming form of the OT problem after being disturbed by the Gumbel noise Eq. (9), which is equivalent to the solution by firstly adding the Gumbel noise, then sorting all items and finally select the top$k$ items. If the perturbed $k$-th and $(k+1)$-th largest items are equal, the algorithm randomly selects one to strictly satisfy the cardinality constraint;

- $\widetilde{\mathbf{T}}^* = \mathrm{Sinkhorn}(\widetilde{\mathbf{D}})$ is the optimal solution of the entropic regularized form of the OT problem after disturbed by the Gumbel noise Eq. (10) solved by Sinkhorn algorithm. It is also the output of our proposed CardNN-GS.

### C.1 Proof of Proposition 2.4

We firstly introduce a Lemma which will be referenced in the proof of Proposition 2.4:

**Lemma C.1.** *Given real numbers $\phi_i, \phi_j$, and $u_i, u_j$ are from i.i.d. $(0,1)$ uniform distribution. After Gumbel perturbation, the probability that $\phi_i + g_\sigma(u_i) > \phi_j + g_\sigma(u_j)$ is:*

$$P(\phi_i + g_\sigma(u_i) > \phi_j + g_\sigma(u_j)) = \frac{1}{1 + \exp{-\frac{\phi_i - \phi_j}{\sigma}}}. \tag{16}$$

*Proof.* Since $g_\sigma(u_i) = -\sigma \log(-\log(u_i))$, $P(\phi_i + g_\sigma(u_i) > \phi_j + g_\sigma(u_j))$ is equivalent to the probability that the following inequality holds:

$$\phi_i - \sigma \log(-\log(u_i)) > \phi_j - \sigma \log(-\log(u_j)) \tag{17}$$

And we have

$$\phi_i - \phi_j > \sigma \log(-\log(u_i)) - \sigma \log(-\log(u_j)) \tag{18}$$

$$\frac{\phi_i - \phi_j}{\sigma} > \log\left(\frac{\log(u_i)}{\log(u_j)}\right) \tag{19}$$

$$e^{\frac{\phi_i - \phi_j}{\sigma}} > \frac{\log(u_i)}{\log(u_j)} \tag{20}$$

Since $u_j \in (0,1)$, $\log(u_j) < 0$. Then we have

$$\log(u_j) < \log(u_i) e^{-\frac{\phi_i - \phi_j}{\sigma}} \tag{21}$$

$$\log(u_j) < \log\left(u_i^{\exp{-\frac{\phi_i - \phi_j}{\sigma}}}\right) \tag{22}$$

$$u_j < u_i^{\exp{-\frac{\phi_i - \phi_j}{\sigma}}} \tag{23}$$

Since $u_i, u_j$ are i.i.d. uniform distributions, the probability when the above formula holds is

$$\int_0^1 \int_0^{u_i^{\exp{-\frac{\phi_i - \phi_j}{\sigma}}}} du_j \, du_i = \int_0^1 u_i^{\exp{-\frac{\phi_i - \phi_j}{\sigma}}} du_i = \frac{1}{1 + \exp{-\frac{\phi_i - \phi_j}{\sigma}}} \tag{24}$$

Thus the probability that $\phi_i + g_\sigma(u_i) > \phi_j + g_\sigma(u_j)$ after Gumbel perturbation is:

$$P(\phi_i + g_\sigma(u_i) > \phi_j + g_\sigma(u_j)) = \frac{1}{1 + \exp{-\frac{\phi_i - \phi_j}{\sigma}}} \tag{25}$$

$\square$

In the following we present the proof of Proposition 2.4:

*Proof of Proposition 2.4.* Recall that we denote $\boldsymbol{\Phi} = [\phi_1, \phi_2, \phi_3, ..., \phi_m]$ as the descending-ordered version of s. By perturbing it with i.i.d. Gumbel noise, we have

$$\widetilde{\boldsymbol{\Phi}} = [\phi_1 + g_\sigma(u_1), \phi_2 + g_\sigma(u_2), \phi_3 + g_\sigma(u_3), ..., \phi_m + g_\sigma(u_m)] \tag{26}$$

where $g_\sigma(u) = -\sigma \log(-\log(u))$ is the Gumbel noise modulated by noise factor $\sigma$, and $u_1, u_2, u_3, ..., u_m$ are i.i.d. uniform distribution. We define $\pi$ as the permutation of sorting $\widetilde{\boldsymbol{\Phi}}$ in descending order, i.e. $\phi_{\pi_1} + g_\sigma(u_{\pi_1}), \phi_{\pi_2} + g_\sigma(u_{\pi_2}), \phi_{\pi_3} + g_\sigma(u_{\pi_3}), ..., \phi_{\pi_m} + g_\sigma(u_{\pi_m})$ are in descending order.

Recall Proposition 2.3, for $\phi_1, \phi_2, \phi_3, ..., \phi_m$ we have

$$\|\mathbf{T}^* - \mathbf{T}^{\tau*}\|_F \leq \frac{2m\tau \log 2}{|\phi_k - \phi_{k+1}|} \tag{27}$$

By substituting $\boldsymbol{\Phi}$ with $\widetilde{\boldsymbol{\Phi}}$ and taking the expectation over $u$, we have

$$\mathbb{E}_u \left[ \|\mathbf{T}^{\sigma*} - \widetilde{\mathbf{T}}^*\|_F \right] \leq \mathbb{E}_u \left[ \frac{2m\tau \log 2}{|\phi_{\pi_k} + g_\sigma(u_{\pi_k}) - \phi_{\pi_{k+1}} - g_\sigma(u_{\pi_{k+1}})|} \right] \tag{28}$$

Based on Lemma C.1, the probability that $\pi_k = i, \pi_{k+1} = j$ is

$$P(\pi_k = i, \pi_{k+1} = j) = \frac{1}{1 + \exp{-\frac{\phi_i - \phi_j}{\sigma}}} \sum_{\forall \pi} \left( \prod_{a=1}^{k-1} \frac{1}{1 + \exp{-\frac{\phi_{\pi_a} - \phi_i}{\sigma}}} \prod_{b=k+2}^{m} \frac{1}{1 + \exp{-\frac{\phi_j - \phi_{\pi_b}}{\sigma}}} \right) \tag{29}$$

where the first term denotes $\phi_i + g_\sigma(u_i) > \phi_j + g_\sigma(u_j)$, the second term denotes all conditions that there are $(k-1)$ items larger than $\phi_i + g_\sigma(u_i)$ and the rest items are smaller than $\phi_j + g_\sigma(u_j)$.

In the following we derive the upper bound of $\mathbb{E}_u \left[ \frac{1}{|\phi_{\pi_k} + g_\sigma(u_{\pi_k}) - \phi_{\pi_{k+1}} - g_\sigma(u_{\pi_{k+1}})|} \right]$. We denote $\mathcal{A}_{i,j}$ as

$$u_i, u_j \in \mathcal{A}_{i,j}, \quad s.t. \quad \phi_i + g_\sigma(u_i) - \phi_j - g_\sigma(u_j) > \epsilon \tag{30}$$

where $\epsilon$ is a sufficiently small number. Then we have

$$\mathbb{E}_u \left[ \frac{1}{|\phi_{\pi_k} + g_\sigma(u_{\pi_k}) - \phi_{\pi_{k+1}} - g_\sigma(u_{\pi_{k+1}})|} \right]$$
$$= \sum_{i \neq j} P(\pi_k = i, \pi_{k+1} = j) \mathbb{E}_{u_i, u_j \in \mathcal{A}_{i,j}} \left[ \frac{1}{|\phi_i + g_\sigma(u_i) - \phi_j - g_\sigma(u_j)|} \right] \tag{31}$$

$$= \sum_{i \neq j} \left( \frac{1}{1 + \exp{-\frac{\phi_i - \phi_j}{\sigma}}} \sum_{\forall \pi} \left( \prod_{a=1}^{k-1} \frac{1}{1 + \exp{-\frac{\phi_{\pi_a} - \phi_i}{\sigma}}} \prod_{b=k+2}^{m} \frac{1}{1 + \exp{-\frac{\phi_j - \phi_{\pi_b}}{\sigma}}} \right. \right.$$
$$\left. \left. \mathbb{E}_{u_i, u_j \in \mathcal{A}_{i,j}} \left[ \frac{1}{|\phi_i + g_\sigma(u_i) - \phi_j - g_\sigma(u_j)|} \right] \right) \right) \tag{32}$$

$$= \sum_{i \neq j} \left( \frac{1}{1 + \exp{-\frac{\phi_i - \phi_j}{\sigma}}} \sum_{\forall \pi} \left( \prod_{a=1}^{k-1} \frac{1}{1 + \exp{-\frac{\phi_{\pi_a} - \phi_i}{\sigma}}} \prod_{b=k+2}^{m} \frac{1}{1 + \exp{-\frac{\phi_j - \phi_{\pi_b}}{\sigma}}} \right. \right.$$
$$\left. \left. \mathbb{E}_{u_i, u_j \in \mathcal{A}_{i,j}} \left[ \frac{1}{|\phi_i - \sigma \log(-\log(u_i)) - \phi_j + \sigma \log(-\log(u_j))|} \right] \right) \right) \tag{33}$$

$$= \sum_{i \neq j} \left( f(\phi_i - \phi_j, \sigma, z) \sum_{\forall \pi} \left( \prod_{a=1}^{k-1} \frac{1}{1 + \exp{-\frac{\phi_{\pi_a} - \phi_i}{\sigma}}} \prod_{b=k+2}^{m} \frac{1}{1 + \exp{-\frac{\phi_j - \phi_{\pi_b}}{\sigma}}} \right) \right) \tag{34}$$

We denote $f(\delta, \sigma, z)$ as:

$$f(\delta, \sigma, z) = \frac{1}{1 + \exp - \frac{\delta}{\sigma}} \mathbb{E}_{u_i, u_j} \left[ \frac{1}{|\delta - \sigma \log(-\log(u_i)) + \sigma \log(-\log(u_j))|} \right] \quad (35)$$
$$s.t. \quad \delta - \sigma \log(-\log(u_i)) + \sigma \log(-\log(u_j)) > z > 0$$

For the probability terms in Eq. (34), for all permutations $\pi$, there must exist $\pi_a, \pi_b$, such that

$$\frac{1}{1 + \exp - \frac{\phi_{\pi_a} - \phi_i}{\sigma}} \leq \frac{1}{1 + \exp - \frac{\phi_k - \phi_i}{\sigma}} \quad (36)$$

$$\frac{1}{1 + \exp - \frac{\phi_j - \phi_{\pi_b}}{\sigma}} \leq \frac{1}{1 + \exp - \frac{\phi_j - \phi_{k+1}}{\sigma}} \quad (37)$$

Thus we have

$$\text{Eq. (34)} \leq \sum_{i \neq j} \left( f(\phi_i - \phi_j, \sigma, z) \frac{1}{1 + \exp - \frac{\phi_k - \phi_i}{\sigma}} \frac{1}{1 + \exp - \frac{\phi_j - \phi_{k+1}}{\sigma}} \right) \quad (38)$$

$$\leq \sum_{i \neq j} \frac{f(\phi_i - \phi_j, \sigma, z)}{(1 + \exp \frac{\phi_i - \phi_k}{\sigma})(1 + \exp \frac{\phi_{k+1} - \phi_j}{\sigma})} \quad (39)$$

By Eq. (16) in Lemma C.1 and substituting $\phi_j - \phi_i$ by $y$, we have

$$\text{Eq. (16)} \Rightarrow P(g_\sigma(u_i) - g_\sigma(u_j) > \phi_j - \phi_i) = \frac{1}{1 + \exp - \frac{\phi_i - \phi_j}{\sigma}} \quad (40)$$

$$\Rightarrow P(g_\sigma(u_i) - g_\sigma(u_j) > y) = \frac{1}{1 + \exp \frac{y}{\sigma}} \quad (41)$$

$$\Rightarrow P(g_\sigma(u_i) - g_\sigma(u_j) < y) = 1 - \frac{1}{1 + \exp \frac{y}{\sigma}} = \frac{1}{1 + \exp - \frac{y}{\sigma}} \quad (42)$$

where the right hand side is exactly the cumulative distribution function (CDF) of standard Logistic distribution by setting $\sigma = 1$:

$$\text{CDF}(y) = \frac{1}{1 + \exp(-y)} \quad (43)$$

Thus $-\log(-\log(u_i)) + \log(-\log(u_j))$ is equivalent to the Logistic distribution whose probability density function (PDF) is

$$\text{PDF}(y) = \frac{d\text{CDF}(y)}{dy} = \frac{1}{\exp(-y) + \exp y + 2} \quad (44)$$

and in this proof we exploit an upper bound of PDF$(y)$:

$$\text{PDF}(y) = \frac{1}{\exp(-y) + \exp y + 2} \leq \frac{1}{y^2 + 4} \quad (45)$$

Based on the Logistic distribution, we can replace $-\sigma \log(-\log(u_i)) + \sigma \log(-\log(u_j))$ by $\sigma y$ where $y$ is from the Logistic distribution. Thus we can derive the upper bound of $f(\delta, \sigma, z)$ as

follows

$$f(\delta, \sigma, z) = \frac{1}{1 + \exp - \frac{\delta}{\sigma}} \cdot \frac{\int_{-\delta/\sigma + z}^{\infty} \frac{1}{\delta + \sigma y} \frac{1}{\exp(-y) + \exp y + 2} dy}{\int_{-\delta/\sigma + z}^{\infty} \frac{1}{\exp(-y) + \exp y + 2} dy} \tag{46}$$

$$= \frac{1}{1 + \exp - \frac{\delta}{\sigma}} \cdot \frac{\int_{-\delta/\sigma + z}^{\infty} \frac{1}{\delta + \sigma y} \frac{1}{\exp(-y) + \exp y + 2} dy}{1 - \frac{1}{1 + \exp(\delta/\sigma - z)}} \tag{47}$$

$$= \frac{1}{1 + \exp - \frac{\delta}{\sigma}} \cdot \frac{\int_{-\delta/\sigma + z}^{\infty} \frac{1}{\delta + \sigma y} \frac{1}{\exp(-y) + \exp y + 2} dy}{\frac{\exp(\delta/\sigma - z)}{1 + \exp(\delta/\sigma - z)}} \tag{48}$$

$$= \frac{1}{1 + \exp - \frac{\delta}{\sigma}} \cdot \frac{\int_{-\delta/\sigma + z}^{\infty} \frac{1}{\delta + \sigma y} \frac{1}{\exp(-y) + \exp y + 2} dy}{\frac{1}{1 + \exp(-\delta/\sigma + z)}} \tag{49}$$

$$= \frac{1 + \exp(-\frac{\delta}{\sigma} + z)}{1 + \exp - \frac{\delta}{\sigma}} \int_{-\delta/\sigma + z}^{\infty} \frac{1}{\delta + \sigma y} \frac{1}{\exp(-y) + \exp y + 2} dy \tag{50}$$

$$\leq \frac{1 + \exp(-\frac{\delta}{\sigma} + z)}{1 + \exp - \frac{\delta}{\sigma}} \int_{-\delta/\sigma + z}^{\infty} \frac{1}{\delta + \sigma y} \frac{1}{y^2 + 4} dy \tag{51}$$

$$= \frac{1 + \exp(-\frac{\delta}{\sigma} + z)}{1 + \exp - \frac{\delta}{\sigma}} \cdot \frac{2\sigma \log\left((z\sigma - \delta)^2 + 4\sigma^2\right) - 2\delta \arctan\left(\frac{z - \delta/\sigma}{2}\right) - 4\sigma \log z + \pi\delta}{4\delta^2 + 16\sigma^2} \tag{52}$$

$$\leq \frac{1 + \exp(-\frac{\delta}{\sigma} + z)}{1 + \exp - \frac{\delta}{\sigma}} \cdot \frac{2\sigma \log\left((z\sigma + |\delta|)^2 + 4\sigma^2\right) - 2\delta \arctan\left(\frac{z - \delta/\sigma}{2}\right) - 4\sigma \log z + \pi\delta}{4\delta^2 + 16\sigma^2} \tag{53}$$

$$= \frac{1 + \exp(-\frac{\delta}{\sigma} + z)}{1 + \exp - \frac{\delta}{\sigma}} \cdot \frac{2\sigma \log\left((z\sigma + |\delta|)^2 + 4\sigma^2\right) - 2\delta \arctan\left(\frac{z - \delta/\sigma}{2}\right) - 2\sigma \log z^2 + \pi\delta}{4\delta^2 + 16\sigma^2} \tag{54}$$

$$= \frac{1 + \exp(-\frac{\delta}{\sigma} + z)}{1 + \exp - \frac{\delta}{\sigma}} \cdot \frac{2\sigma \log\left(\frac{(z\sigma + |\delta|)^2 + 4\sigma^2}{z^2}\right) - 2\delta \arctan\left(\frac{z - \delta/\sigma}{2}\right) + \pi\delta}{4\delta^2 + 16\sigma^2} \tag{55}$$

$$\leq \frac{1 + \exp(-\frac{\delta}{\sigma} + z)}{1 + \exp - \frac{\delta}{\sigma}} \cdot \frac{2\sigma \log\left(\frac{(z\sigma + |\delta| + 2\sigma)^2}{z^2}\right) - 2\delta \arctan\left(\frac{z - \delta/\sigma}{2}\right) + \pi\delta}{4\delta^2 + 16\sigma^2} \tag{56}$$

$$= \frac{1 + \exp(-\frac{\delta}{\sigma} + z)}{1 + \exp - \frac{\delta}{\sigma}} \cdot \frac{4\sigma \log\left(\frac{z\sigma + |\delta| + 2\sigma}{z}\right) - 2\delta \arctan\left(\frac{z - \delta/\sigma}{2}\right) + \pi\delta}{4\delta^2 + 16\sigma^2} \tag{57}$$

$$= \frac{1 + \exp(-\frac{\delta}{\sigma} + z)}{1 + \exp - \frac{\delta}{\sigma}} \cdot \frac{4\sigma \log\left(\frac{z\sigma + |\delta| + 2\sigma}{z}\right) + \delta\left(\pi - 2\arctan\left(\frac{z - \delta/\sigma}{2}\right)\right)}{4\delta^2 + 16\sigma^2} \tag{58}$$

$$\leq \frac{1 + \exp(-\frac{\delta}{\sigma} + z)}{1 + \exp - \frac{\delta}{\sigma}} \cdot \frac{4\sigma \log\left(\frac{z\sigma + |\delta| + 2\sigma}{z}\right) + |\delta|\left(\pi - 2\arctan\left(\frac{z - \delta/\sigma}{2}\right)\right)}{4\delta^2 + 16\sigma^2} \tag{59}$$

$$\leq \frac{1 + \exp(-\frac{\delta}{\sigma} + z)}{1 + \exp - \frac{\delta}{\sigma}} \cdot \frac{4\sigma \log\left(\frac{z\sigma + |\delta| + 2\sigma}{z}\right) + |\delta|\left(\pi - 2\arctan\left(-\frac{\delta}{2\sigma}\right)\right)}{4\delta^2 + 16\sigma^2} \tag{60}$$

$$= \frac{1 + \exp(-\frac{\delta}{\sigma} + z)}{1 + \exp - \frac{\delta}{\sigma}} \cdot \frac{4\sigma \log\left(\frac{z\sigma + |\delta| + 2\sigma}{z}\right) + |\delta|\left(\pi + 2\arctan\left(\frac{\delta}{2\sigma}\right)\right)}{4\delta^2 + 16\sigma^2} \tag{61}$$

where Eq. (51) is because $\frac{1}{\exp(-y)+\exp y+2} \leq \frac{1}{y^2+4}$, and Eq. (59) is because $\pi - 2\arctan(\frac{z-\delta/\sigma}{2}) \geq 0$. With probability at least $(1-\epsilon)$, we have

$$z = \log\frac{1+\epsilon\exp\frac{\delta}{\sigma}}{1-\epsilon} \geq -\log(1-\epsilon) \tag{62}$$

$$\frac{1+\exp\left(-\frac{\delta}{\sigma}+z\right)}{1+\exp-\frac{\delta}{\sigma}} = \frac{1}{1-\epsilon} \tag{63}$$

Thus

$$f(\delta,\sigma,z) \leq \text{Eq. (61)} = \frac{1}{1-\epsilon}\frac{4\sigma\log\left(\frac{z\sigma+|\delta|+2\sigma}{z}\right)+|\delta|\left(\pi+2\arctan\left(\frac{\delta}{2\sigma}\right)\right)}{4\delta^2+16\sigma^2} \tag{64}$$

$$\leq\frac{1}{1-\epsilon}\frac{4\sigma\log\left(\sigma-\frac{|\delta|+2\sigma}{\log(1-\epsilon)}\right)+|\delta|\left(\pi+2\arctan\left(\frac{\delta}{2\sigma}\right)\right)}{4\delta^2+16\sigma^2} \tag{65}$$

Thus we have

$$\text{Eq. (39)} \leq \sum_{i\neq j}\left(\frac{4\sigma\log\left(\sigma-\frac{|\phi_i-\phi_j|+2\sigma}{\log(1-\epsilon)}\right)+|\phi_i-\phi_j|\left(\pi+2\arctan\left(\frac{\phi_i-\phi_j}{2\sigma}\right)\right)}{(1-\epsilon)(4(\phi_i-\phi_j)^2+16\sigma^2)(1+\exp\frac{\phi_i-\phi_k}{\sigma})(1+\exp\frac{\phi_{k+1}-\phi_j}{\sigma})}\right) \tag{66}$$

In conclusion, with probability at least $(1-\epsilon)$, we have

$$\mathbb{E}_u\left[\|\mathbf{T}^{\sigma*}-\widetilde{\mathbf{T}}^*\|_F\right] \leq \sum_{i\neq j}\frac{(2\log 2)m\tau\left(4\sigma\log\left(\sigma-\frac{|\phi_i-\phi_j|+2\sigma}{\log(1-\epsilon)}\right)+|\phi_i-\phi_j|\left(\pi+2\arctan\frac{\phi_i-\phi_j}{2\sigma}\right)\right)}{(1-\epsilon)(4(\phi_i-\phi_j)^2+16\sigma^2)(1+\exp\frac{\phi_i-\phi_k}{\sigma})(1+\exp\frac{\phi_{k+1}-\phi_j}{\sigma})} \tag{67}$$

$$=\sum_{i\neq j}\frac{(\log 2)m\tau\left(2\sigma\log\left(\sigma-\frac{|\phi_i-\phi_j|+2\sigma}{\log(1-\epsilon)}\right)+|\phi_i-\phi_j|\left(\frac{\pi}{2}+\arctan\frac{\phi_i-\phi_j}{2\sigma}\right)\right)}{(1-\epsilon)((\phi_i-\phi_j)^2+4\sigma^2)(1+\exp\frac{\phi_i-\phi_k}{\sigma})(1+\exp\frac{\phi_{k+1}-\phi_j}{\sigma})} \tag{68}$$

$$=(\log 2)m\tau\sum_{i\neq j}\Omega(\phi_i,\phi_j,\sigma,\epsilon) \tag{69}$$

And we denote $\Omega(\phi_i,\phi_j,\sigma,\epsilon)$ as

$$\Omega(\phi_i,\phi_j,\sigma,\epsilon) = \frac{2\sigma\log\left(\sigma-\frac{|\phi_i-\phi_j|+2\sigma}{\log(1-\epsilon)}\right)+|\phi_i-\phi_j|\left(\frac{\pi}{2}+\arctan\frac{\phi_i-\phi_j}{2\sigma}\right)}{(1-\epsilon)((\phi_i-\phi_j)^2+4\sigma^2)(1+\exp\frac{\phi_i-\phi_k}{\sigma})(1+\exp\frac{\phi_{k+1}-\phi_j}{\sigma})} \tag{70}$$

$\square$

## C.2 PROOF OF COROLLARY 2.5

Corollary 2.5 is the simplified version of Proposition 2.4 by studying the dominant components.

*Proof.* For $\Omega(\phi_i,\phi_j,\sigma,\epsilon)$ in Proposition 2.4, we have

$$\Omega(\phi_i,\phi_j,\sigma,\epsilon) \leq \frac{2\sigma\log\left(\sigma-\frac{|\phi_i-\phi_j|+2\sigma}{\log(1-\epsilon)}\right)+|\phi_i-\phi_j|\left(\frac{\pi}{2}+\arctan\frac{\phi_i-\phi_j}{2\sigma}\right)}{(1-\epsilon)((\phi_i-\phi_j)^2+4\sigma^2)} \tag{71}$$

$$\leq\frac{2\sigma\log\left(\sigma-\frac{|\phi_i-\phi_j|+2\sigma}{\log(1-\epsilon)}\right)+|\phi_i-\phi_j|\pi}{(1-\epsilon)((\phi_i-\phi_j)^2+4\sigma^2)} \tag{72}$$

$$=O\left(\frac{\sigma\log(\sigma+|\phi_i-\phi_j|)+|\phi_i-\phi_j|}{(\phi_i-\phi_j)^2+\sigma^2}\right) \tag{73}$$

$$=\widetilde{O}\left(\frac{\sigma+|\phi_i-\phi_j|}{(\phi_i-\phi_j)^2+\sigma^2}\right) \tag{74}$$

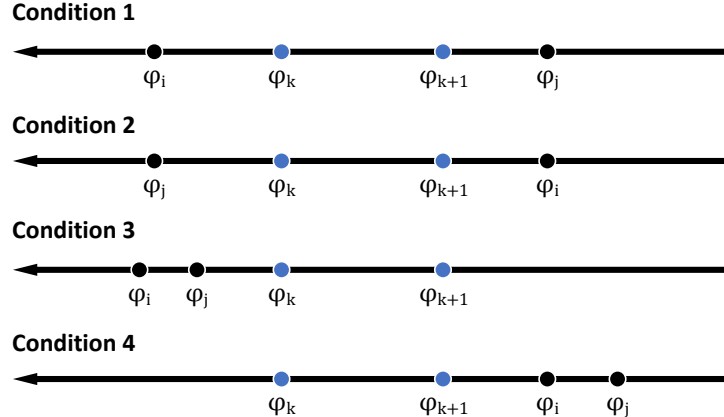

Figure 7: Four conditions are considered in our proof. It is worth noting that $\phi_i, \phi_j$ must not lie between $\phi_k, \phi_{k+1}$, because we define $\phi_k, \phi_{k+1}$ as two adjacent items in the original sorted list.

where we regard $(1 - \epsilon)$ as a constant (i.e. assuming high probability), and $\widetilde{O}(\cdot)$ means ignoring the logarithm terms.

Then we have

$$\mathbb{E}_u \left[ \|\mathbf{T}^{\sigma*} - \widetilde{\mathbf{T}}^*\|_F \right] \leq (\log 2) m\tau \sum_{i \neq j} \widetilde{O} \left( \frac{\sigma + |\phi_i - \phi_j|}{(\phi_i - \phi_j)^2 + \sigma^2} \right) \tag{75}$$

$$= (\log 2) m\tau \widetilde{O} \left( \frac{\sigma + |\phi_i - \phi_j|}{(\phi_i - \phi_j)^2 + \sigma^2} \right)_{\forall i \neq j} \tag{76}$$

$$= \widetilde{O} \left( \frac{m\tau(\sigma + |\phi_i - \phi_j|)}{(\phi_i - \phi_j)^2 + \sigma^2} \right)_{\forall i \neq j} \tag{77}$$

$\square$

## C.3 PROOF AND REMARKS ON COROLLARY 2.6

In the following, we prove Corollary 2.6 and add some remarks about the relationship between the Sinkhorn and the Gumbel-Sinkhorn methods: the Sinkhorn method (CardNN-S) is a special case of the Gumbel-Sinkhorn method (CardNN-GS) when we set $\sigma \to 0^+$. To more formally address Corollary 2.6, we have the following proposition:

**Proposition C.2.** *Assume the values of $\phi_k, \phi_{k+1}$ are unique[1], under probability at least $(1 - \epsilon)$, we have*

$$\lim_{\sigma \to 0^+} \mathbb{E}_u \left[ \|\mathbf{T}^{\sigma*} - \widetilde{\mathbf{T}}^*\|_F \right] \leq \frac{(\pi \log 2) m\tau}{(1 - \epsilon)|\phi_k - \phi_{k+1}|} \tag{78}$$

which differs from the conclusion of Proposition 2.3 by only a constant factor.

*Proof.* Since $\sigma \to 0^+$, the first term in $\Omega(\phi_i, \phi_j, \sigma, \epsilon)$'s numerator becomes 0. For the second term, we discuss four conditions as shown in Fig. 7, except for the following condition: $\phi_i = \phi_k$, $\phi_j = \phi_{k+1}$.

---

[1]For a compact proof, we make this assumption that the values of $\phi_k, \phi_{k+1}$ are unique. If there are duplicate values of $\phi_k, \phi_{k+1}$, the bound only differs by a constant multiplier, therefore, does not affect our conclusion: Sinkhorn method (CardNN-S) is a special case of the Gumbel-Sinkhorn method (CardNN-GS) when $\sigma \to 0^+$.

**Condition 1**. If $\phi_i \geq \phi_k, \phi_j \leq \phi_{k+1}$ (equalities do not hold at the same time), we have at least $\phi_i - \phi_k > 0$ or $\phi_{k+1} - \phi_j > 0$. Then we have

$$\lim_{\sigma \to 0^+} \frac{1}{(1 + \exp \frac{\phi_i - \phi_k}{\sigma})(1 + \exp \frac{\phi_{k+1} - \phi_j}{\sigma})} = 0 \tag{79}$$

$$\Rightarrow \lim_{\sigma \to 0^+} \Omega(\phi_i, \phi_j, \sigma, \epsilon) = 0 \tag{80}$$

**Condition 2**. For any case that $\phi_i < \phi_j$, we have $\phi_i - \phi_j < 0$, thus

$$\lim_{\sigma \to 0^+} \arctan \frac{\phi_i - \phi_j}{\sigma} = -\frac{\pi}{2} \tag{81}$$

$$\Rightarrow \lim_{\sigma \to 0^+} \frac{\pi}{2} + \arctan \frac{\phi_i - \phi_j}{\sigma} = 0 \tag{82}$$

$$\Rightarrow \lim_{\sigma \to 0^+} \Omega(\phi_i, \phi_j, \sigma, \epsilon) = 0 \tag{83}$$

**Condition 3**. If $\phi_i \geq \phi_j \geq \phi_k$ (equalities do not hold at the same time), we have $\phi_i - \phi_k > 0$. Then we have

$$\lim_{\sigma \to 0^+} \frac{1}{1 + \exp \frac{\phi_i - \phi_k}{\sigma}} = 0 \tag{84}$$

$$\Rightarrow \lim_{\sigma \to 0^+} \Omega(\phi_i, \phi_j, \sigma, \epsilon) = 0 \tag{85}$$

**Condition 4**. If $\phi_{k+1} \geq \phi_i \geq \phi_j$ (equalities do not hold at the same time), we have $\phi_{k+1} - \phi_j > 0$. Then we have

$$\lim_{\sigma \to 0^+} \frac{1}{1 + \exp \frac{\phi_{k+1} - \phi_j}{\sigma}} = 0 \tag{86}$$

$$\Rightarrow \lim_{\sigma \to 0^+} \Omega(\phi_i, \phi_j, \sigma, \epsilon) = 0 \tag{87}$$

Therefore, if $\phi_i \neq \phi_k$ and $\phi_j \neq \phi_{k+1}$, the second term $\Omega(\phi_i, \phi_j, \sigma, \epsilon)$ degenerates to 0 when $\sigma \to 0^+$. Thus we have the following conclusion by only considering $\phi_i = \phi_k, \phi_j = \phi_{k+1}$:

$$\lim_{\sigma \to 0^+} \mathbb{E}_u \left[ \|\mathbf{T}^{\sigma*} - \widetilde{\mathbf{T}}^*\|_F \right] \leq \frac{(\log 2)m\tau \left( |\phi_k - \phi_{k+1}| \left( \frac{\pi}{2} + \arctan \frac{\phi_k - \phi_{k+1}}{2\sigma} \right) \right)}{(1 - \epsilon)(\phi_k - \phi_{k+1})^2} \tag{88}$$

$$\leq \frac{(\pi \log 2)m\tau}{(1 - \epsilon)|\phi_k - \phi_{k+1}|} \tag{89}$$

$$\square$$

**Remarks**. Based on the above conclusion, if $|\phi_k - \phi_{k+1}| > 0$, with $\sigma \to 0^+$, Eq. (11) degenerates to the bound in Eq. (6) and only differs by a constant factor:

$$\lim_{\sigma \to 0^+} \mathbb{E}_u \left[ \|\mathbf{T}^{\sigma*} - \widetilde{\mathbf{T}}^*\|_F \right] \leq \frac{(\pi \log 2)m\tau}{(1 - \epsilon)|\phi_k - \phi_{k+1}|} \tag{90}$$

where a strong assumption that $|\phi_k - \phi_{k+1}| > 0$ is made, and the bound diverges if $\phi_k = \phi_{k+1}$. Since $\phi_k, \phi_{k+1}$ are predictions by a neural network, such an assumption may not be satisfied. In comparison, given $\sigma > 0$, the conclusion with Gumbel noise in Eq. (11) is bounded for any $\phi_k, \phi_{k+1}$. The strength of the theoretical results is also validated in experiment (see Tables 5 and 4), including the homotopy version CardNN-HGS.

## D  ALGORITHM DETAILS FOR SOLVING DETERMINISTIC CO PROBLEMS

Due to limited pages, we do not include detailed algorithm blocks on how to solve deterministic CO problems in the main paper. Here we elaborate on our implementation for solving facility location problem (FLP) in Alg. 2, and max covering problem (MCP) in Alg. 3.

---

**Algorithm 2: CardNN-GS/HGS for Solving the Facility Location Problem**

---

**Input:** the distance matrix $\mathbf{\Delta}$; learning rate $\alpha$; softmax temperature $\beta$; CardNN-GS parameters $k, \tau, \sigma, \#\mathrm{G}$.

1 **if** *Training* **then**
2     Randomly initialize neural network weights $\theta$;
3 **if** *Inference* **then**
4     Load pretrained neural network weights $\theta$; $J_{best} = +\infty$;
5 **while** *not converged* **do**
6     $\mathbf{s} = \mathrm{SplineCNN}_\theta(\mathbf{\Delta})$; $[\widetilde{\mathbf{T}}_1, \widetilde{\mathbf{T}}_2, ..., \widetilde{\mathbf{T}}_{\#\mathrm{G}}] = \mathrm{CardNN\text{-}GS}(\mathbf{s}, k, \tau, \sigma, \#\mathrm{G})$;
7     for all $i$, $\widetilde{J}_i = \mathrm{sum}(\mathrm{softmax}(-\beta\mathbf{\Delta} \circ \widetilde{\mathbf{T}}_i[2,:]) \circ \mathbf{\Delta})$;
8     $J = \mathrm{mean}([\widetilde{J}_1, \widetilde{J}_2, ..., \widetilde{J}_{\#\mathrm{G}}])$;
9     **if** *Training* **then**
10       update $\theta$ with respect to the gradient $\frac{\partial J}{\partial \theta}$ and learning rate $\alpha$ by gradient descend;
11     **if** *Inference* **then**
12       update $\mathbf{s}$ with respect to the gradient $\frac{\partial J}{\partial \mathbf{s}}$ and learning rate $\alpha$ by gradient descend;
13       for all $i$, $\widetilde{J}_i = \mathrm{sum}(\min(\mathbf{\Delta} \circ \mathrm{TopK}(\widetilde{\mathbf{T}}_i[2,:]^\top)))$; $J_{best} = \min([\widetilde{J}_1, \widetilde{J}_2, ..., \widetilde{J}_{\#\mathrm{G}}], J_{best})$;
14 **if** *Homotopy Inference* **then**
15     Shrink the value of $\tau$ and jump to line 5;

**Output:** Learned network weights $\theta$ (if training)/The best objective $J_{best}$ (if inference).

---

**Algorithm 3: CardNN-GS/HGS for Solving the Max Covering Problem**

---

**Input:** bipartite adjacency $\mathbf{A}$; values $\mathbf{v}$; learning rate $\alpha$; CardNN-GS parameters $k, \tau, \sigma, \#\mathrm{G}$.

1 **if** *Training* **then**
2     Randomly initialize neural network weights $\theta$;
3 **if** *Inference* **then**
4     Load pretrained neural network weights $\theta$; $J_{best} = 0$;
5 **while** *not converged* **do**
6     $\mathbf{s} = \mathrm{GraphSage}_\theta(\mathbf{A})$; $[\widetilde{\mathbf{T}}_1, \widetilde{\mathbf{T}}_2, ..., \widetilde{\mathbf{T}}_{\#\mathrm{G}}] = \mathrm{CardNN\text{-}GS}(\mathbf{s}, k, \tau, \sigma, \#\mathrm{G})$;
7     for all $i$, $\widetilde{J}_i = \min(\widetilde{\mathbf{T}}_i[2,:]\mathbf{A}, 1)^\top \cdot \mathbf{v}$; $J = \mathrm{mean}([\widetilde{J}_1, \widetilde{J}_2, ..., \widetilde{J}_{\#\mathrm{G}}])$;
8     **if** *Training* **then**
9       update $\theta$ with respect to the gradient $\frac{\partial J}{\partial \theta}$ and learning rate $\alpha$ by gradient ascent;
10     **if** *Inference* **then**
11       update $\mathbf{s}$ with respect to the gradient $\frac{\partial J}{\partial \mathbf{s}}$ and learning rate $\alpha$ by gradient ascent;
12       for all $i$, $\widetilde{J}_i = (\mathrm{TopK}(\widetilde{\mathbf{T}}_i[2,:])\mathbf{A})^\top \cdot \mathbf{v}$; $J_{best} = \max([\widetilde{J}_1, \widetilde{J}_2, ..., \widetilde{J}_{\#\mathrm{G}}], J_{best})$;
13 **if** *Homotopy Inference* **then**
14     Shrink the value of $\tau$ and jump to line 5;

**Output:** Learned network weights $\theta$ (if training)/The best objective $J_{best}$ (if inference).

---

## E    MORE DETAILS ABOUT DETERMINISTIC CO EXPERIMENT

### E.1    DATASET DETAILS

**The Starbucks location dataset for FLP**. This dataset is built based on the project named Starbucks Location Worldwide 2021 version[2], which is scraped from the open-accessible Starbucks store locator webpage[3]. We analyze and select 4 cities with more than 100 Starbucks stores, which are London (166 stores), New York City (260 stores), Shanghai (510 stores), and Seoul (569 stores). The locations considered are the real locations represented as latitude and longitude. For simplic-

---

[2]https://www.kaggle.com/datasets/kukuroo3/starbucks-locations-worldwide-2021-version
[3]https://www.starbucks.com/store-locator

Table 4: Objective score ↓, optimal gap ↓ and inference time (in seconds) ↓ comparison of the facility location problem, including mean and standard deviation computed from all test instances. The problem is to select $k$ facilities from $m$ locations.

| EGN/CardNN are CO networks | k=30, m=500 | | | k=50, m=800 | | |
|---|---|---|---|---|---|---|
| | objective ↓ | optimal gap ↓ | time ↓ (sec) | objective ↓ | optimal gap ↓ | time ↓ (sec) |
| greedy | 2.841±0.093 | 0.167±0.026 | 1.771±0.017 | 2.671±0.066 | 0.168±0.018 | 4.779±0.035 |
| SCIP 7.0 (t=120s/200s) | 4.470±1.918 | 0.348±0.295 | 118.068±48.055 | 5.258±1.018 | 0.552±0.146 | 243.919±54.118 |
| Gurobi 9.0 (t=120s/200s) | 2.453±0.142 | 0.033±0.042 | 125.589±0.606 | 3.364±0.268 | 0.335±0.055 | 214.360±3.785 |
| Gurobi 9.0 (optimal) | 2.365±0.063 | 0.000±0.000 | 314.798±116.858 | 2.221±0.041 | 0.000±0.000 | 648.213±194.486 |
| EGN (efficient) | 3.032±0.195 | 0.217±0.048 | 0.830±0.308 | 2.879±0.155 | 0.226±0.039 | 0.988±0.140 |
| EGN (accurate) | 2.795±0.140 | 0.152±0.035 | 123.559±12.278 | 2.697±0.116 | 0.175±0.031 | 191.091±13.141 |
| CardNN-S (Sec. 2.1) | 2.753±0.154 | 0.139±0.041 | 7.127±1.241 | 2.462±0.079 | 0.097±0.023 | 6.427±1.050 |
| CardNN-GS (Sec. 2.2) | 2.420±0.072 | 0.023±0.009 | 76.534±6.321 | 2.283±0.050 | 0.027±0.008 | 120.689±2.405 |
| CardNN-HGS (Sec. 2.2) | **2.416±0.073** | 0.021±0.009 | 103.742±4.778 | **2.275±0.048** | 0.023±0.007 | 158.400±3.498 |

ity, we do not consider the real-world distances between any two stores; instead, we test with both Euclidean distance and Manhattan distance. We set k=30, and the objective values reported are distances ×100km.

**The Twitch dataset for MCP**. This social network dataset is collected by Rozemberczki et al. (2021) and the edges represent the mutual friendships between streamers. The streamers are categorized by their streaming language, resulting in 6 social networks for 6 languages. The social networks are DE (9498 nodes), ENGB (7126 nodes), ES (4648 nodes), FR (6549 nodes), PTBR (1912 nodes), and RU (4385 nodes). The objective is to cover more viewers, measured by the sum of the logarithmic number of viewers. We took the logarithm to enforce diversity because those top streamers usually have the dominant number of viewers. We set k=50.

### E.2   IMPLEMENTATION DETAILS

Our algorithms are implemented by PyTorch and the graph neural network modules are based on Fey & Lenssen (2019). In our paper, we optimize the hyperparameters by greedy search on a small subset of problem instances (~5) and set the best configuration of hyperparameters for CardNN-S/GS/HGS. The hyperparameters of EGN (Karalias & Loukas, 2020) are tuned in the same way. Here are the hyperparameters used to reproduce our experiment results:

- For the **Max Covering Problem (MCP)**, we empirically set the learning rate $\alpha = 0.1$. For the hyperparameters of CardNN, we have $\tau = 0.05, \sigma = 0.15$ for CardNN-GS, $\tau = 0.05$ for CardNN-S, and $\tau = (0.05, 0.04, 0.03), \sigma = 0.15$ for the Homotopy version CardNN-HGS. We set $\#G = 1000$ samples for CardNN-GS and CardNN-HGS.

- For the **Facility Location Problem (FLP)**, we set the learning rate $\alpha = 0.1$. For the hyperparameters of CardNN, we have $\tau = 0.05, \sigma = 0.25$ for CardNN-GS, $\tau = 0.05$ for CardNN-S, and we set $\tau = (0.05, 0.04, 0.03), \sigma = 0.25$ for the Homotopy version CardNN-HGS. We set $\#G = 500$ samples for CardNN-GS and CardNN-HGS. The softmax temperature for facility location is empirically set as twice of the cardinality constraint: $T = 100$ if $k = 50$, $T = 60$ if $k = 30$.

- For the **Predictive Portfolio Optimization Problem**, we set the learning rate $\alpha = 10^{-3}$. For our CardNN-GS module, we set $\tau = 0.05, \sigma = 0.1$, and set the Gumbel samples as $\#G = 1000$. During inference, among all 1000 portfolio predictions, we return the best portfolio found based on the predicted prices, and we empirically find such a strategy beneficial for finding better portfolios on the real test set.

All experiments are done on a workstation with i7-9700K@3.60GHz CPU, 16GB memory, and RTX2080Ti GPU.

### E.3   DETAILED EXPERIMENT RESULTS

In the main paper, we only plot the experiment results on both synthetic datasets and real-world datasets due to limited pages. In Table 4 and 5, we report the digits from the synthetic experiments, which are in line with Fig. 3.

Table 5: Objective score ↑, gap ↓, and inference time (in seconds) ↓ of max covering. Under cardinality constraint $k$, the problem is to select from $m$ sets to cover a fraction of $n$ objects. For the gray entry, the Gurobi solver fails to return the optimal solution within 24 hours, thus reported as out-of-time.

| EGN/CardNN are CO networks | k=50, m=500, n=1000 | | | k=100, m=1000, n=2000 | | |
|---|---|---|---|---|---|---|
| | objective ↑ | gap ↓ | time ↓ (sec) | objective ↑ | gap ↓ | time ↓ (sec) |
| greedy | 44312.8±818.4 | 0.011±0.007 | **0.024±0.000** | 88698.9±1217.5 | 0.008±0.004 | **0.089±0.001** |
| SCIP 7.0 (t=100s/120s) | 43497.4±875.6 | 0.029±0.011 | 100.136±0.097 | 86269.9±1256.3 | 0.035±0.006 | 120.105±0.498 |
| Gurobi 9.0 (t=100s/120s) | 43937.2±791.5 | 0.019±0.008 | 100.171±0.085 | 86862.1±1630.5 | 0.028±0.011 | 120.277±0.139 |
| Gurobi 9.0 (optimal) | OOT | OOT | OOT | OOT | OOT | OOT |
| EGN (efficient) | 37141.4±896.0 | 0.171±0.015 | 0.244±0.107 | 74633.7±1449.6 | 0.165±0.010 | 0.525±0.229 |
| EGN (accurate) | 39025.2±791.9 | 0.129±0.008 | 40.542±4.056 | 77488.9±1088.2 | 0.133±0.006 | 93.670±8.797 |
| CardNN-S (Sec. 2.1) | 42034.9±773.1 | 0.062±0.008 | 4.935±1.167 | 83289.0±1331.0 | 0.068±0.007 | 5.368±1.014 |
| CardNN-GS (Sec. 2.2) | 44710.3±770.9 | **0.002±0.002** | 28.104±0.465 | 89264.8±1232.1 | 0.001±0.002 | 60.685±0.045 |
| CardNN-HGS (Sec. 2.2) | **44723.9±763.2** | **0.002±0.002** | 39.575±0.595 | **89340.8±1221.6** | **0.000±0.001** | 89.764±0.128 |

**Some remarks about EGN on real-world dataset**. Since the sizes of our real-world problems are relatively small, we mainly adopt a transfer learning setting: the CO networks are firstly trained on the synthetic data, and then tested on the corresponding real-world datasets. All our CardNN models follow this setting. However, the transfer learning ability of EGN seems less satisfying, and we empirically find the performance of EGN degenerates significantly when transferred to a different dataset. In Fig. 4, we exploit the advantage of self-supervised learning for EGN: we allow EGN to be trained in a self-supervised manner on the real-world dataset. To avoid the scatter plots looking too sparse, we ignore the training time cost when plotting Fig. 4 since it does not affect our main conclusion (performance rank: CardNN-HGS > CardNN-GS > CardNN-S > EGN).

We list the detailed experiment results on real-world problems in Tables 6-19.

Table 6: FLP-Starbucks London dataset (Euclidean distance)

| m=166, k=30 | objective↓ | time (sec)↓ |
|---|---|---|
| **Greedy** | 0.047 | 0.6 |
| **SCIP 7.0 (t=60s)** | 0.040 | 2.8 |
| **Gurobi 9.0 (t=60s)** | 0.040 | 7.2 |
| **Gurobi 9.0 (optimal)** | 0.040 | 7.2 |
| **EGN (train on synthetic)** | 0.171 | 0.2 |
| **EGN-accu (train on synthetic)** | 0.171 | 25.6 |
| **EGN (train on test)** | 0.080 | 0.1 |
| **EGN-accu (train on test)** | 0.078 | 17.2 |
| **CardNN-S** | 0.054 | 8.2 |
| **CardNN-GS** | 0.042 | 19.8 |
| **CardNN-HGS** | 0.042 | 50.3 |

Table 7: FLP-Starbucks NewYork dataset (Euclidean distance)

| m=260, k=30 | objective↓ | time (sec)↓ |
|---|---|---|
| **Greedy** | 0.033 | 0.9 |
| **SCIP 7.0 (t=60s)** | 0.028 | 16.5 |
| **Gurobi 9.0 (t=60s)** | 0.028 | 60.6 |
| **Gurobi 9.0 (optimal)** | 0.028 | 126.5 |
| **EGN (train on synthetic)** | 0.174 | 0.1 |
| **EGN-accu (train on synthetic)** | 0.174 | 26.0 |
| **EGN (train on test)** | 0.089 | 0.1 |
| **EGN-accu (train on test)** | 0.057 | 27.0 |
| **CardNN-S** | 0.174 | 2.3 |
| **CardNN-GS** | 0.030 | 20.2 |
| **CardNN-HGS** | 0.029 | 50.8 |

Table 8: FLP-Starbucks Shanghai dataset (Euclidean distance)

| m=510, k=30 | objective↓ | time (sec)↓ |
|---|---|---|
| **Greedy** | 0.172 | 1.9 |
| **SCIP 7.0 (t=60s)** | 10.484 | 106.1 |
| **Gurobi 9.0 (t=60s)** | 0.222 | 62.3 |
| **Gurobi 9.0 (optimal)** | 0.139 | 313.1 |
| **EGN (train on synthetic)** | 1.561 | 0.3 |
| **EGN-accu (train on synthetic)** | 1.561 | 58.5 |
| **EGN (train on test)** | 0.360 | 0.3 |
| **EGN-accu (train on test)** | 0.360 | 56.5 |
| **CardNN-S** | 0.165 | 9.0 |
| **CardNN-GS** | 0.162 | 20.7 |
| **CardNN-HGS** | 0.155 | 51.3 |

Table 9: FLP-Starbucks Seoul dataset (Euclidean distance)

| m=569, k=30 | objective↓ | time (sec)↓ |
|---|---|---|
| **Greedy** | 0.245 | 2.1 |
| **SCIP 7.0 (t=60s)** | 14.530 | 145.2 |
| **Gurobi 9.0 (t=60s)** | 0.424 | 62.9 |
| **Gurobi 9.0 (optimal)** | 0.188 | 540.8 |
| **EGN (train on synthetic)** | 2.680 | 0.3 |
| **EGN-accu (train on synthetic)** | 2.680 | 57.9 |
| **EGN (train on test)** | 0.497 | 0.3 |
| **EGN-accu (train on test)** | 0.497 | 63.7 |
| **CardNN-S** | 0.373 | 9.0 |
| **CardNN-GS** | 0.284 | 21.8 |
| **CardNN-HGS** | 0.212 | 52.7 |

Table 10: FLP-Starbucks London dataset (Manhattan distance)

| m=166, k=30 | objective↓ | time (sec)↓ |
|---|---|---|
| **Greedy** | 2.441 | 0.5 |
| **SCIP 7.0 (t=60s)** | 2.390 | 2.9 |
| **Gurobi 9.0 (t=60s)** | 2.390 | 2.2 |
| **Gurobi 9.0 (optimal)** | 2.390 | 2.2 |
| **EGN (train on synthetic)** | 4.793 | 0.2 |
| **EGN-accu (train on synthetic)** | 4.793 | 19.4 |
| **EGN (train on test)** | 3.210 | 0.1 |
| **EGN-accu (train on test)** | 3.008 | 18.4 |
| **CardNN-S** | 2.688 | 4.4 |
| **CardNN-GS** | 2.457 | 20.6 |
| **CardNN-HGS** | 2.424 | 51.6 |

Table 11: FLP-Starbucks NewYork dataset (Manhattan distance)

| m=260, k=30 | objective↓ | time (sec)↓ |
|---|---|---|
| **Greedy** | 2.734 | 0.9 |
| **SCIP 7.0 (t=60s)** | 2.565 | 9.6 |
| **Gurobi 9.0 (t=60s)** | 2.565 | 22.9 |
| **Gurobi 9.0 (optimal)** | 2.565 | 22.8 |
| **EGN (train on synthetic)** | 4.066 | 0.2 |
| **EGN-accu (train on synthetic)** | 4.066 | 30.6 |
| **EGN (train on test)** | 3.998 | 0.1 |
| **EGN-accu (train on test)** | 3.500 | 27.5 |
| **CardNN-S** | 4.066 | 2.4 |
| **CardNN-GS** | 2.898 | 15.2 |
| **CardNN-HGS** | 2.845 | 30.1 |

Table 12: FLP-Starbucks Shanghai dataset (Manhattan distance)

| m=510, k=30 | objective↓ | time (sec)↓ |
|---|---|---|
| **Greedy** | 9.024 | 1.8 |
| **SCIP 7.0 (t=60s)** | 59.626 | 101.8 |
| **Gurobi 9.0 (t=60s)** | 8.931 | 62.2 |
| **Gurobi 9.0 (optimal)** | 8.439 | 201.7 |
| **EGN (train on synthetic)** | 21.566 | 0.3 |
| **EGN-accu (train on synthetic)** | 21.566 | 55.2 |
| **EGN (train on test)** | 17.951 | 0.3 |
| **EGN-accu (train on test)** | 11.601 | 53.0 |
| **CardNN-S** | 10.780 | 5.1 |
| **CardNN-GS** | 8.784 | 26.0 |
| **CardNN-HGS** | 8.774 | 58.8 |

Table 13: FLP-Starbucks Seoul dataset (Manhattan distance)

| m=569, k=30 | objective↓ | time (sec)↓ |
|---|---|---|
| **Greedy** | 10.681 | 2.0 |
| **SCIP 7.0 (t=60s)** | 83.952 | 95.2 |
| **Gurobi 9.0 (t=60s)** | 14.579 | 65.7 |
| **Gurobi 9.0 (optimal)** | 9.911 | 335.7 |
| **EGN (train on synthetic)** | 18.206 | 0.3 |
| **EGN-accu (train on synthetic)** | 18.206 | 56.6 |
| **EGN (train on test)** | 15.168 | 0.3 |
| **EGN-accu (train on test)** | 15.069 | 64.6 |
| **CardNN-S** | 13.154 | 5.1 |
| **CardNN-GS** | 10.146 | 28.6 |
| **CardNN-HGS** | 10.003 | 63.2 |

Table 14: MCP-Twitch DE dataset

| m=n=9498, k=50 | objective↑ | time (sec)↓ |
|---|---|---|
| Greedy | 51452 | 0.3 |
| SCIP 7.0 (optimal) | 51481 | 1.5 |
| Gurobi 9.0 (optimal) | 51481 | 5.8 |
| EGN (train on synthetic) | 850 | 15.5 |
| EGN-accu (train on synthetic) | 11732 | 303.0 |
| EGN (train on test) | 43036 | 15.4 |
| EGN-accu (train on test) | 43069 | 303.2 |
| CardNN-S | 51478 | 16.0 |
| CardNN-GS | 51481 | 28.7 |
| CardNN-HGS | 51481 | 56.1 |

Table 15: MCP-Twitch ENGB dataset

| m=n=7126, k=50 | objective↑ | time (sec)↓ |
|---|---|---|
| Greedy | 26748 | 0.1 |
| SCIP 7.0 (optimal) | 26757 | 0.3 |
| Gurobi 9.0 (optimal) | 26757 | 0.8 |
| EGN (train on synthetic) | 5066 | 7.6 |
| EGN-accu (train on synthetic) | 7749 | 147.6 |
| EGN (train on test) | 18725 | 7.5 |
| EGN-accu (train on test) | 19296 | 147.2 |
| CardNN-S | 26745 | 15.6 |
| CardNN-GS | 26757 | 21.7 |
| CardNN-HGS | 26757 | 42.6 |

Table 16: MCP-Twitch ES dataset

| m=n=4648, k=50 | objective↑ | time (sec)↓ |
|---|---|---|
| Greedy | 25492 | 0.1 |
| SCIP 7.0 (optimal) | 25492 | 0.3 |
| Gurobi 9.0 (optimal) | 25492 | 1.1 |
| EGN (train on synthetic) | 1183 | 3.1 |
| EGN-accu (train on synthetic) | 1489 | 57.6 |
| EGN (train on test) | 17612 | 3.1 |
| EGN-accu (train on test) | 17872 | 58.0 |
| CardNN-S | 25492 | 15.7 |
| CardNN-GS | 25492 | 12.6 |
| CardNN-HGS | 25492 | 24.9 |

Table 17: MCP-Twitch FR dataset

| m=n=6549, k=50 | objective↑ | time (sec)↓ |
|---|---|---|
| Greedy | 39665 | 0.2 |
| SCIP 7.0 (optimal) | 39694 | 1.2 |
| Gurobi 9.0 (optimal) | 39694 | 10.7 |
| EGN (train on synthetic) | 21508 | 6.2 |
| EGN-accu (train on synthetic) | 21701 | 119.2 |
| EGN (train on test) | 32439 | 6.2 |
| EGN-accu (train on test) | 32533 | 119.4 |
| CardNN-S | 39687 | 15.8 |
| CardNN-GS | 39693 | 19.9 |
| CardNN-HGS | 39694 | 39.1 |

Table 18: MCP-Twitch PTBR dataset

| m=n=1912, k=50 | objective↑ | time (sec)↓ |
|---|---|---|
| **Greedy** | 14141 | 0.0 |
| **SCIP 7.0 (optimal)** | 14163 | 0.1 |
| **Gurobi 9.0 (optimal)** | 14163 | 0.3 |
| **EGN (train on synthetic)** | 1402 | 0.8 |
| **EGN-accu (train on synthetic)** | 7329 | 14.4 |
| **EGN (train on test)** | 10173 | 0.3 |
| **EGN-accu (train on test)** | 10173 | 5.3 |
| **CardNN-S** | 14155 | 15.6 |
| **CardNN-GS** | 14163 | 10.1 |
| **CardNN-HGS** | 14163 | 19.7 |

Table 19: MCP-Twitch RU dataset

| m=n=4385, k=50 | objective↑ | time (sec)↓ |
|---|---|---|
| **Greedy** | 25755 | 0.1 |
| **SCIP 7.0 (optimal)** | 25778 | 0.2 |
| **Gurobi 9.0 (optimal)** | 25778 | 0.5 |
| **EGN (train on synthetic)** | 7762 | 2.7 |
| **EGN-accu (train on synthetic)** | 7917 | 51.5 |
| **EGN (train on test)** | 18148 | 2.6 |
| **EGN-accu (train on test)** | 18156 | 48.3 |
| **CardNN-S** | 25776 | 15.9 |
| **CardNN-GS** | 25778 | 11.9 |
| **CardNN-HGS** | 25778 | 23.6 |

### E.4 ABLATION STUDY ON HYPERPARAMETERS

Firstly, we want to add some remarks about the selection of hyperparameters:

- $\#G$ (**number of Gumbel samples**): $\#G$ affects how many samples are taken during training and inference for CardNN-GS. A larger $\#G$ (i.e. more samples) will be more appealing, because CardNN-GS will have a lower variance when estimating the objective score, and it will have a higher probability of discovering better solutions. However, $\#G$ cannot be arbitrarily large because the GPU has limited memory, also it is harmful to the efficiency if $\#G$ is too large. In experiments, we set an adequate $\#G$ (e.g. $\#G = 1000$) and ensure that it can fit into the GPU memory of our workstation (2080Ti, 11G).

- $\tau$ (**entropic regularization factor of Sinkhorn**): Theoretically, $\tau$ controls the gap of the continuous Sinkhorn solution to the discrete solution, and a smaller $\tau$ will lead to a tightened gap. This property is validated by our theoretical findings in Proposition 2.4. Unfortunately, $\tau$ cannot be arbitrarily small, because a smaller $\tau$ requires more Sinkhorn iterations to converge. Besides, a smaller $\tau$ means the algorithm being closer to the discrete version, and the gradient will be more likely to explode. Therefore, given a fixed number of Sinkhorn iterations (100) to ensure the efficiency of our algorithm, we need trial-and-error to discover the suitable $\tau$ for both CardNN-S and CardNN-GS. The grid search results below show that our selection of $\tau$ fairly balances the performances of both CardNN-S and CardNN-GS.

- $\sigma$ (**Gumbel noise factor**): As derived in Proposition 2.4, a larger $\sigma$ is beneficial for a tightened constraint-violation term. However, it is also worth noting that $\sigma$ cannot be arbitrarily large because our theoretical derivation only considers the expectation but not the variance. A larger $\sigma$ means a larger variance, demanding a larger number of samples and bringing computational and memory burdens. In the experiments, we first determine a $\tau$, and then find a suitable $\sigma$ by greedy search on a small subset ($\sim$5) of problem instances.

We conduct an ablation study about the sensitivity of hyperparameters by performing an extensive grid search near the configuration used in our max covering experiments ($\tau = 0.05, \sigma = 0.15, \#G = 1000$). We choose the k=50, m=500, n=1000 max covering problem, and we have the following results for CardNN-GS and CardNN-S (higher is better):

Table 20: Ablation study result of CardNN-GS with $\#G = 1000$.

| $\sigma =$ \ $\tau =$ | 0.01 | 0.05 | 0.1 |
|---|---|---|---|
| 0.1 | 42513.4 | 44759.2 | **45039.5** |
| 0.15 | 41456.5 | 44710.3 | 44837.2 |
| 0.2 | 41264.3 | 44638.1 | 44748.2 |

Table 21: Ablation study result of CardNN-GS with $\#G = 800$.

| $\sigma =$ \ $\tau =$ | 0.01 | 0.05 | 0.1 |
|---|---|---|---|
| 0.1 | 42511.6 | 44754.6 | **45037.6** |
| 0.15 | 41421.4 | 44705.8 | 44841.5 |
| 0.2 | 41235.9 | 44651.5 | 44748.6 |

Table 22: Ablation study result of CardNN-S.

| $\tau =$ | 0.001 | 0.005 | 0.01 | 0.05 | 0.1 |
|---|---|---|---|---|---|
| objective score | 35956.6 | 42013.3 | **42520.8** | 42034.9 | 40721.2 |

Under the configuration used in our paper, both CardNN-S and CardNN-GS have relatively good results. Our grid search result shows that our CardNN-GS is not very sensitive to $\sigma$ if we have $\tau = 0.05$ or $0.1$, and the result of $\tau = 0.01$ is inferior because the Sinkhorn algorithm may not converge. The results of $\#G = 1000$ are all better than $\#G = 800$, suggesting that a larger $\#G$ is appealing if we have enough GPU memory. It is also discovered that CardNN-S seems to be able to accept a smaller value of $\tau$ compared to CardNN-GS, possibly because adding the Gumbel noise will increase the divergence of elements thus performs in a sense similar to decreasing $\tau$ when considering the convergence of Sinkhorn.

## F  DETAILS OF PREDICTIVE PORTFOLIO OPTIMIZATION

Some details of the portfolio optimization model is omitted due to limited pages. Here we elaborate on the entire process of doing portfolio optimization under the "pred-*and*-opt" paradigm, with LSTM and our CardNN-GS.

**Training steps**:

1. Denote the index of "now" as $t = 0$. $\{p_t|t < 0\}$ means the percentage change of prices of each day in history, $\{p_t|t \geq 0\}$ means the percentage change of prices of each day in future.

2. An encoder-decoder LSTM module predicts the prices in the future:

$$\{p_t|t \geq 0\}, \mathbf{h} = \text{LSTM}(\{p_t|t < 0\}),$$

where $\mathbf{h}$ denotes the hidden state of LSTM.

3. Compute risk and return for the future:

$$\mu = \text{mean}(\{p_t|t \geq 0\}), \mathbf{\Sigma} = \text{cov}(\{p_t|t \geq 0\}).$$

4. In the CardNN-GS module, predict $\mathbf{s}$ (the probability of selected each asset) from $\mathbf{h}$:

$$\mathbf{s} = \text{fully-connected}(\mathbf{h}).$$

5. Enforce the cardinality constraint by Gumbel-Sinkhorn layer introduced in Sec 3.2, whereby there are $\#G$ Gumbel samples:

$$\{\widetilde{\mathbf{T}}_i | i = 1, 2, ..., \#G\} = \text{Gumbel-Sinkhorn}(\mathbf{s})$$

6. Compute the weights of each asset based on the second row of $\widetilde{\mathbf{T}}_i$ ($r_f$ is risk-free return, set as 3%):

$$\mathbf{x}_i = \mathbf{\Sigma}^{-1}(\mu - r_f), \mathbf{x}_i = \text{relu}(\mathbf{x}_i \odot \widetilde{\mathbf{T}}_i[2, :]), \mathbf{x}_i = \mathbf{x}_i / \text{sum}(\mathbf{x})$$

7. Based on the ground-truth prices in the future $\{p_t^{gt} | t \geq 0\}$, compute the ground truth risk and return:

$$\mu^{gt} = \text{mean}(\{p_t^{gt} | t \geq 0\}), \mathbf{\Sigma}^{gt} = \text{cov}(\{p_t^{gt} | t \geq 0\}).$$

8. Estimate the ground-truth Sharpe ratio in the future, if we invest based on $\mathbf{x}_i$:

$$\widetilde{J}_i = \frac{(\mu^{gt} - r_f)^\top \mathbf{x}_i}{\sqrt{\mathbf{x}_i^\top \mathbf{\Sigma}^{gt} \mathbf{x}_i}}.$$

9. The self-supervised loss is the average over all Gumbel samples:

$$Loss = -\text{mean}(\widetilde{J}_1, \widetilde{J}_2, \widetilde{J}_3, ..., \widetilde{J}_{\#G})$$

**Testing steps**:

Follow training steps 1-6 to predict $\mu, \mathbf{\Sigma}, \{\mathbf{x}_i | i = 1, 2, ..., \#G\}$.

7. Estimate the predicted Sharpe ratio in the future, if we invest based on $\mathbf{x}_i$:

$$\widetilde{J}_i = \frac{(\mu - r_f)^\top \mathbf{x}_i}{\sqrt{\mathbf{x}_i^\top \mathbf{\Sigma} \mathbf{x}_i}}.$$

8. Return $\mathbf{x}_{best} = \mathbf{x}_i$ with the highest $\widetilde{J}_i$ and enforce hard cardinality constraint on $\mathbf{x}_{best}$ by hard top$k$.

9. Evaluate based on the ground-truth Sharpe ratio:

$$J = \frac{(\mu^{gt} - r_f)^\top \mathbf{x}_{best}}{\sqrt{\mathbf{x}_{best}^\top \mathbf{\Sigma}^{gt} \mathbf{x}_{best}}}.$$

## G  VISUALIZATION OF MORE PORTFOLIOS

In Fig. 8, we provide more visualizations of the portfolios predicted by our "predict-*and*-optimize" CardNN pipeline (blue), the traditional "predict-*then*-optimize" pipeline based on LSTM and Gurobi (orange), and the historical-data based "history-opt" (purple). In general, portfolio optimization means a trade-off between risks and returns, and we can draw an efficient frontier where the portfolios on this frontier are the Pareto optimal for risks and returns, i.e. for a portfolio on the efficient frontier, one cannot achieve higher returns unless s/he could accept higher risks. Being closer to the efficient frontier means a portfolio is better. Besides, it is also worth noting that reaching the efficient frontier is nearly infeasible in predictive portfolio optimization because our predictions of future asset prices are always with errors.

## H  DETAILS ON USING EXISTING ASSETS

The following open-source resources are used in this paper and we sincerely thank the authors and contributors for their great work.

- **Implementation of Erdos Goes Neural**. Paper: Karalias & Loukas (2020). URL: https://github.com/Stalence/erdos_neu. No open-source license is found on the GitHub webpage.

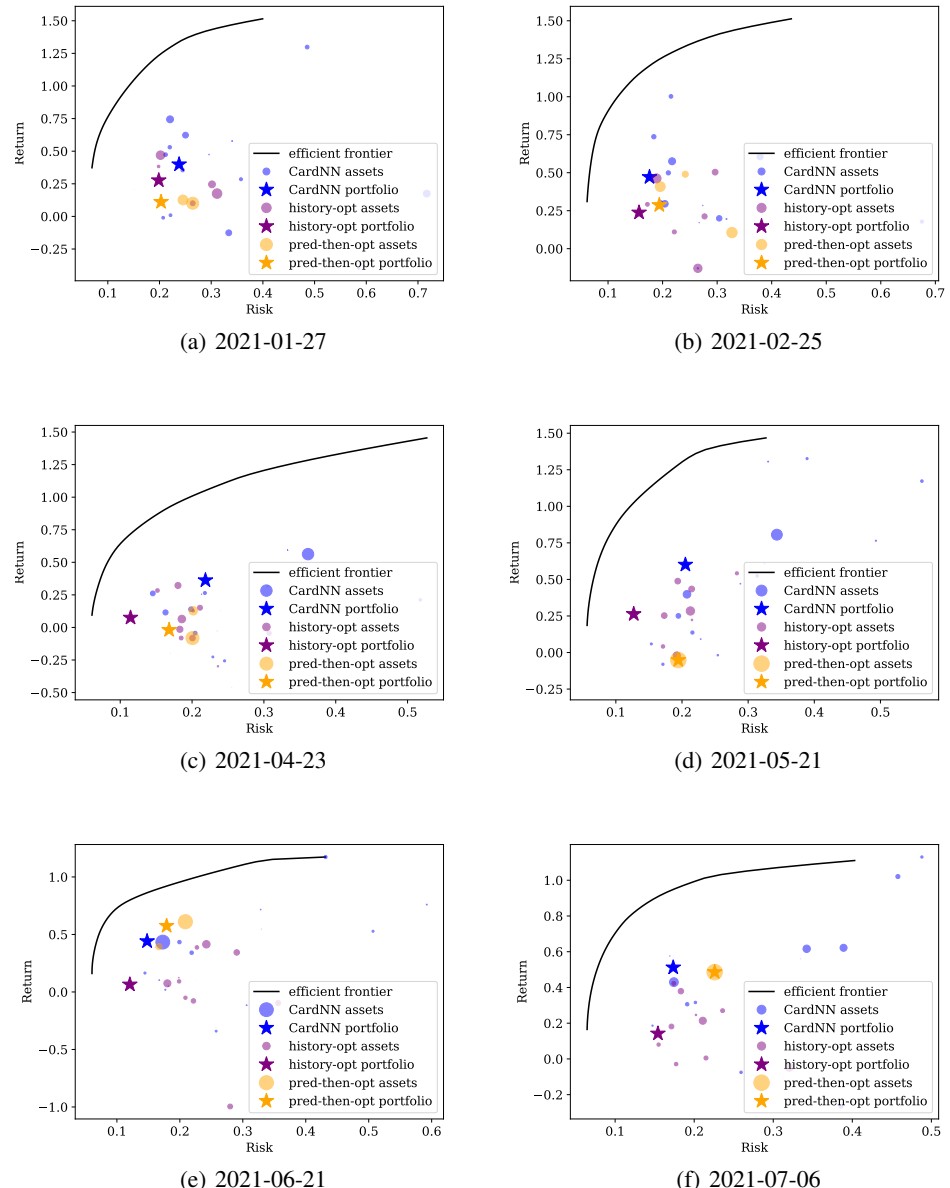

Figure 8: Visualization of predicted portfolios. The labels denote the starting dates of the portfolios.

- **SCIP solver**. Paper: Gamrath et al. (2020). URL: `https://scip.zib.de/`. ZIB Academic License.

- **ORLIB**. Paper: Beasley (1990). URL: `http://people.brunel.ac.uk/~mastjjb/jeb/orlib/scpinfo.html`. MIT License.

- **Starbucks Locations Worldwide (2021 version)**. URL: `https://www.kaggle.com/datasets/kukuroo3/starbucks-locations-worldwide-2021-version`. CC0: Public Domain License.

- **Twitch Social Networks (from MUSAE project)**. Paper: Rozemberczki et al. (2021). Project URL: `https://github.com/benedekrozemberczki/MUSAE` Data

URL: [http://snap.stanford.edu/data/twitch-social-networks.html](http://snap.stanford.edu/data/twitch-social-networks.html). GPL-3.0 License.

And we are also using the Gurobi commercial solver under academic license. See details about Gurobi's academic license at [https://www.gurobi.com/academia/academic-program-and-licenses/](https://www.gurobi.com/academia/academic-program-and-licenses/).

