# OpenReview forum: "Towards One-shot Neural Combinatorial Solvers: Theoretical and Empirical Notes on the Cardinality-Constrained Case"
_ICLR.cc/2023/Conference — ICLR 2023 poster_

### Official Review · Reviewer_oRCe · 2022-10-21

**Confidence:** 3
**Correctness:** 4
**Technical Novelty And Significance:** 3
**Empirical Novelty And Significance:** 2
**Recommendation:** 6

**Clarity, Quality, Novelty And Reproducibility:**

- The paper is well written and engaging (at least to me!!).
- The Gumbel-Sinkhorn approach to cardinality constraints makes sense, and as far as I know has not been tried before.
- Paper contains some details on the implementation, and the authors promise that "Source code and pretrained models will be made publicly available.".

**Strength And Weaknesses:**

**Strengths**

- A good amount of technical novelty. The OT with Gumbel noise approach is interesting to me and is an attractive wya to recast the top-k selection problem.
- the paper is well written in my opinion.
- I am satisfied with the empirical results---a number of problems are considered including facility location, finding max covers, and a portfolio optimization problem. I am not convinced that portfolio optimization really needs cardinality constraints in practice (but definitely a whole bunch of other constraints!), but as a proof of concept I am willing to accept the setup.

**Weaknesses:**

- I am a little concerned at the limited impact of the approach. Cardinality constraints are fairly specific, and I cannot foresee too many people needing to use this method. I am, however, strongly against rejection based on impact. I just believe it important to at least raise the point.
- The method still only softly enforces the constraint. Perhaps some more discussion explaining why hard enforcement is a difficult problem would help myself and other readers understand why we should feel satisfied with soft enforcement.

**Summary Of The Paper:**

This paper considers how to build a cardinality constraint into neural network architectures being trained to solve combinatorial optimization tasks. This work builds on the Erdos Goes Neural approach of Karalias & Loukas, which uses a loss penalization term to enforce constrains. While Karalias & Loukas do provide high probability guarantees on constraint satisfaction for the penalization approach, in practice it requires careful selection of the weighting of the penalization loss.

This paper takes a different approach, and design a differentiable output layer based on solving a Sinkhorn problem with added Gumbel noise. Whilst still only softly enforcing the constraint, the source of the relaxation is very different from Karalias & Loukas since it is not in the loss function itself. Empirical results suggest that in practice the approach doers a pretty good job at enforcing the constraint.

**Summary Of The Review:**

Congratulations on a nice paper, I enjoyed reading it. The differentiable k subset selection approach is novel and interesting to me. I am also very much in favor of further exploration of neural combinatorial optimization approaches, and I think this paper plays a role in deepening the literature on this subject. In all I am broadly positive in my recommendation.


Some questions and comments:

- While I agree that the naive top-k selection will not work as a training approach, I am not sure it is actually because of non-differentiability. The function in question is a map $f$ from the probability simplex to the family of k-cardinality sets. This function is differentiable almost everywhere, besides the case where the k and k+1 probability are equal. I think the real reason why this $f$ does not suit gradient based training is that it is piecewise constant, and so gradient based training will not see any useful gradient directions with which to update the model. I would like to ask the authors to consider this and either explain to me any misconception I have, or to adjust the motivation given in the paper accordingly.
- Prop 3.4 is a high probability result. What is the randomness over? The LHS already takes an expectation over the Gumbel variable so I am not sure what other randomness it left.

---

> ### Author Response · Authors · 2022-11-10
> **Response to Reviewer oRCe (2/2)**
>
> **Q3: More discussion explaining why hard enforcement (of cardinality constraint) is a difficult problem.**
> * We are actually open to any new methods that can offer approximated gradients through the cardinality constraint. The perturbation-based differentiation methods [9,10,11] could enforce hard constraints while still being able to compute the gradient, and they have the advantage of guaranteeing a zero constraint violation. However, our study in Table 2 shows that the learned performances of hard-constrained methods [9,10,11] are inferior to softly constrained methods (CardNN-S & CardNN-GS).
>
>   We paste Table 2 here for your easy reference (max covering score, higher is better):
>
>   | CardNN+[9] | CardNN+[10] | CardNN+[11] | CardNN-S (ours) | CardNN-GS (ours) |
>   | ---------- | ----------- | ----------- | --------------- | ---------------- |
>   | 32499.7    | 37618.9     | 38899.6     | 41001.3         | 44710.3          |
>
>   Looking at the topic of differentiating through piece-wise constant functions, one challenge is that there seems no golden standard for judging whether an approximated gradient is good or not. On one hand, the gradient approximation should, to some degree, reflect the original gradient. On the other hand, the approximation cannot be too close to the original because the original gradient is too sparse and cannot be utilized by neural networks.
>
>   A possible explanation for the success of soft approximation is that its input-output mapping is more smooth, making it easier for the network to converge. We also empirically discover that the gradient through Sinkhorn and Gumbel-Sinkhorn is denser than the gradient through hard constraint methods [9,10,11].
>
> **Q4: Naïve top-k is differentiable. The only issue is that its gradient could not be utilized.**
> * We agree with the reviewer on this point. As we wrote on Page 2, "One technical issue is that for a discrete-output network (CO as a special case) whose input-output mappings are piece-wise constant, the real gradient is 0 almost everywhere and infinite at the boundaries where the output changes. Such a gradient cannot guide the training of neural networks". We correct the "non-differentiable" expression on Page 2 accordingly.
>
> **Q5: Prop 3.4 is a high probability result. What is the randomness over?**
> * According to our proof in Appendix, the high-probability term comes from Eq (61) where we cannot take the integration directly and we exploit the high probability of $|(\phi_i + g_i) - (\phi_j + g_j)| \neq 0$ ($\phi_i, \phi_j$ are inputs to the OT layer, $g_i, g_j$ are two i.i.d. Gumbel distributions).
>
>   An intuitive explanation is that after being disturbed by Gumbel noise, it is still possible (though of very low probability) that $\phi_i+g_i$ will be equal to $\phi_j + g_j$. If $\phi_i+g_i=\phi_j + g_j$, according to Prop 3.3, the constraint violation becomes infinite, and the derived bound turns diverged. Such a diverging case is pruned by introducing the high-probability term.
>
> Please feel free to raise any further questions and we are looking forward to hearing your feedback!
>
> **Reference**
>
> [1] T-J Chang, Nigel Meade, John E Beasley, and Yazid M Sharaiha. Heuristics for cardinality constrained portfolio optimization. Computers & Operations Research, 27(13):1271–1302, 2000
>
> [2] Zalan Borsos, Mojmır Mutny, and Andreas Krause. Coresets via bilevel optimization for continual learning and streaming. Neural Info. Process. Systems, 2020.
>
> [3] Abhimanyu Dubey, Moitreya Chatterjee, and Narendra Ahuja. Coreset-based neural network compression. In Eur. Conf. Comput. Vis., September 2018
>
> [4] Lingxiao Huang, Shaofeng Jiang, and Nisheeth Vishnoi. Coresets for clustering with fairness constraints. Neural Info. Process. Systems, 2019.
>
> [5] David Arthur and Sergei Vassilvitskii. K-means++: The advantages of careful seeding. ACM-SIAM Symposium on Discrete Algorithms,  2007.
>
> [6] Elias Khalil, Hanjun Dai, Yuyu Zhang, Bistra Dilkina, and Le Song. Learning combinatorial optimization algorithms over graphs. In Neural Info. Process. Systems, 2017.
>
> [7] O. Vinyals, M. Fortunato, and N. Jaitly, Pointer networks. Neural Info. Process. Systems, 2015.
>
> [8] Fudong Wang, Nan Xue, Jin-Gang Yu, and Gui-Song Xia. Zero-assignment constraint for graph matching with outliers. Comp. Vis. Pattern Recog., 2020.
>
> [9] Marin Vlastelica Pogancic, Anselm Paulus, Vit Musil, Georg Martius, and Michal Rolinek. Differentiation of blackbox combinatorial solvers. In Int. Conf. Learn. Rep., 2019
>
> [10] Quentin Berthet, Mathieu Blondel, Olivier Teboul, Marco Cuturi, Jean-Philippe Vert, and Francis Bach. Learning with differentiable perturbed optimizers. Neural Info. Process. Systems, 33:9508–9519, 2020
>
> [11] Brandon Amos, Vladlen Koltun, and J. Zico Kolter. The Limited Multi-Label Projection Layer. arXiv preprint arXiv:1906.08707, 2019.

---

> ### Author Response · Authors · 2022-11-10
> **Response to Reviewer oRCe (1/2)**
>
> We would like to express our sincere gratitude to the reviewer for recognizing the novelty of our paper. Here we set out below our response to your questions:
>
> >***Q1: About the cardinality constraint and other constraints in portfolio optimization.***
> * For portfolio optimization, we follow the seminal work [1] which introduces the cardinality constraint. The cardinality constraint is adopted for maintaining a limited number of assets which can also save operational costs. We also agree with the reviewer that there are other important constraints in portfolio optimization that are worth exploring in future work.
>
> >***Q2: About the impact of cardinality constraints.***
> * We sincerely appreciate your constructive ideas about the impact of cardinality constraints. Here we try to introduce more background and motivation as follows.
>
>   **Cardinality, especially hard enforcement of cardinality is widely practical and useful.** The cardinality constraint itself is a well-established area with a long-standing and broad impact on machine learning, theoretical computer science, operation research, and finance (just to name a few).
>
>   For example, the hard cardinality constraint can be treated as a special case of the coreset problem which has been intensively studied in recent machine learning and vision conferences [2,3,4] which is in fact currently towards the between of theoretical computer science [5] and machine learning. Our work can also handle soft cardinality (i.e. the general coreset) because the Gumbel-Sinkhorn is a continuous relaxation.
>
>   Given the fact that existing cardinality constraint solvers are mostly learning-free, we strongly believe our neural solver with bounded constraint violation can be of great value to both academia and industry whereby the cardinality constraint needs to be satisfied.
>
>   **The proposed paradigm can incorporate other constraints beyond cardinality.** As stated above, we originally think cardinality itself is of great importance to deserve a full research paper. While in fact, our end-to-end one-shot CO learning and solving paradigm could further incorporate other types of constraints.
>
>   When preparing this conference paper, our intention was not to distract the theme of this paper to other problems so we decided to focus on cardinality. Yet it is indeed feasible to extend our work to other types of constraints. We add the permutation constraint as an example here as encoded by Sinkhorn and optimal transport (OT). We are a bit refrained from elaborating on this point in the main paper as it needs to involve more comparison and discussion on the large body of permutation optimization problems and approaches. But we would be happy to add more details and discussions if the reviewers felt that they are needed.
>
>   The permutation constraint can be directly applied to popular problem instances like TSP and graph matching under our one-shot learning and problem-solving paradigm. Compared with the mainstream auto-regressive scheme in learning for solving TSP (e.g. S2V-DQN [6] and pointer net [7]), conceptually, our one-shot model can be more efficient. We report our preliminary results on using our paradigm for solving TSP on synthetic graph data of size 20, 50, and 100 as the mean of 10000 trials as follows:
>
>   | optimal gap     | TSP20 | TSP50  | TSP100 |
>   | --------------- | ----- | ------ | ------ |
>   | Sinkhorn        | 0.27% | 0.95%  | 2.48%  |
>   | S2V-DQN [6]     | 1.42% | 5.16%  | 7.03%  |
>   | pointer-net [7] | 1.15% | 34.48% | -      |
>
>   One step further, we are also exploring a unified Sinkhorn network to tackle both permutation and cardinality constraints, whereby its application could be found in graph matching with outliers in both graphs. Our preliminary result on Pascal VOC dataset (with outliers) is as follows:
>   | graph matching with outliers             | accuracy |
>   | ---------------------------------------- | ------------- |
>   | Sinkhorn net for permutation+cardinality | 60.4          |
>   | Sinkhorn net for permutation             | 58.8          |
>   | ZACR solver [8]                          | 30.2          |
>
>
>   The above results are preliminary proof-of-concepts, and we believe there is room for further improvement. If you feel needed, we are happy to elaborate on the above results and technical details in the main paper (may need to reorganize the paper due to space limitations).

---

### Official Review · Reviewer_m4Wt · 2022-10-23

**Confidence:** 2
**Correctness:** 3
**Technical Novelty And Significance:** 3
**Empirical Novelty And Significance:** 2
**Recommendation:** 6

**Clarity, Quality, Novelty And Reproducibility:**

Clarity

The submission is hard to follow to me. First, the submission is not self-contained, with some components deferred to existing works not specified. Second, some notations are not well defined or not defined when first used.

Novelty

The proposed methodology seems to be novel.

Reproducibility

The experiment setup of the submission is provided in the submission.

**Strength And Weaknesses:**

Strength

1. The motivation of the submission is clear. CO is a challenging and important problem. It is a motivating question to solve CO while forcing the constraints are satisfied.

2. The empirical performance of the proposed method is thoroughly studied in three different tasks.

Weaknesses

1. The notations of the submission are confusing. Many notations first appear without any definitions. For example the notations in figure 1 appear without definitions. I cannot find the definition of $i$ anywhere in the submission. The definitions of $\mu^{gt}$ and $\Sigma^{gt}$ are also missing. Then, there are notions like $[2,:]$ and $sum(softmax(...))$ which are not like math equations but codes.

2. Many concepts in the submission appear without or with very little definition or explanation. For example, the problem encoder is a key component of the method, but first appears in the figure 1, simply described as a model to predict $\bm{s}$. However, the definition of $\bm{s}$, how it connects to the CO, and why we need to estimate a probability like this is not provided, making the submission not self-contained

It would be great if authors can clarify these confusions.

3. Would it be possible for authors to explain why the CO problem is equivalent to an optimal transport problem on $\bm{s}$? It seems that this is taken for granted without further explanation.

4. Does the gradient flow to the problem encoder part? If so, in the portfolio optimization task, this means that the the gradient flows to the price prediction part. As a result, the stock prediction may learn a prediction not so good at prediction but just to make the CO objective higher. But in the portfolio optimization, the problem setup is to fix the $\mu$ and $\sigma$ to as close as possible to the ground-truth ones.

5. For the portfolio optimization, a yearly return as 40% seems to be overly high in practice. Would it be possible for authors to provide some intuitions behind such a high number. Also, when calculating the efficient frontier, I am wondering what is the return and volatility used. What is  $\mu^{gt}$ and $\Sigma^{gt}$? I also find this application not very suitable, since it does not matter whether the cardinality constraint is absolutely satisfied or just "mostly" satisfied in portfolio optimization.


**Summary Of The Paper:**

The submission studies combinatorial optimization (CO) with cardinality constraints by neural networks. Different from the methods incorporating the constraints as penalties in to the objective function, the submission incorporates the constraints into the network architecture such that these constraints are guaranteed to be satisfied. The performance of the proposed method is demonstrated in several tasks including the facility location problem, max covering problem and portfolio optimization.

**Summary Of The Review:**

The submission considers a challenging and motivational problem. The proposed solution is novel and thoroughly demonstrated on three different tasks. However, for now, there are too many confusing part for me to evaluate the correctness and contribution of the methodology.

---

> ### Author Response · Authors · 2022-11-10
> **Response to Reviewer m4Wt (2/2)**
>
> >***Q3: Would it be possible for authors to explain why the CO problem is equivalent to an optimal transport problem on $\mathbf{s}$?***
> * Sorry for leaving you the impression that there is an equivalence though we did not mention that and it is NOT. The OT layer is developed to enforce cardinality constraints after the problem encoder network. It is the problem encoder network that learns how to solve CO problems, just like other CO networks e.g. [2].
>
> >***Q4.1: Does the gradient flow to the problem encoder part?***
> * Yes. The gradient flows to the price prediction part.
>
> >***Q4.2: In portfolio optimization, the stock prediction may learn a prediction not so good at prediction but just to make the CO objective higher. But in the portfolio optimization, the problem setup is to fix the $\mu$ and $\Sigma$ to be as close as possible to the ground-truth ones.***
> * Thanks for your comments. We are not sure if we understand your comments correctly since these two sentences both refer to portfolio optimization and it makes us a bit confused. It would be appreciated if you could clarify it.
>
>   In our practical setting, we did not aim for prediction accuracy yet the loss refers to the final Sharpe ratio objective. Your mentioned "fixing $\mu$ and $\Sigma$" way of portfolio optimization corresponds to the "pred-*then*-opt" baseline discussed above and can be found in Table 3. The price prediction of "pred-*then*-opt" is more accurate, but its Sharpe ratio is inferior to our "pred-*and*-opt" method. A possible explanation is that *when making investment decisions, predicting the future trend is usually more important than predicting the exact prices*, being aware that the price prediction always contains errors. Our "pred-*and*-opt" method, to some degree, learns the future trend by directly learning over the Sharpe ratio.
>
>   We paste Table 3 here for your easy reference:
>
>   | Methods         | predictor+optimizer | prediction MSE $\downarrow$ | Sharpe ratio $\uparrow$ |
>   | --------------- | ------------------- | --------------------------- | ----------------------- |
>   | history-opt     | none+Gurobi         | (no prediction)             | 0.697                   |
>   | pred-*then*-opt | LSTM+Gurobi         | **0.153**                   | 1.116                   |
>   | pred-*and*-opt  | LSTM+CardNN-GS      | 1.382                       | **2.146**               |
>
> >***Q5.1: For portfolio optimization, a yearly return as 40% seems to be overly high in practice. Would it be possible for authors to provide some intuitions behind such a high number.***
> * Our experiments are performed in the year 2021 when the stock market is in great prosperity (the whole market value, measured by S&P 500, grows by 23.5% in 2021). Many hedge funds even surpass 40% return in 2021, for example, Haidar Jupiter Composite reaches 69.52% net return, according to public information available on Google. Frankly speaking, our model probably will not keep such a high return in the year 2022.
>
> >***Q5.2: When calculating the efficient frontier, I am wondering what is the return and volatility used.***
> * The ground truth prices in 2021 are used to compute return and volatility, and further plot the efficient frontier. The efficient frontier does not have the cardinality constraint. Also, as we mention on Page 9: "Note that reaching the efficient frontier is nearly impossible as the prediction always contains errors".
>
> >***Q5.3: Can the cardinality constraint be "mostly" satisfied in portofolio optimization?***
> * For portfolio optimization, we follow the seminal work [3] which introduces the cardinality constraint. The cardinality constraint is adopted for maintaining a limited number of assets which can also save operational costs. We partially agree with the reviewer that a "mostly" cardinality-constrained portfolio may be suitable, and our approach could smoothly fit into this setting because our treatment of the cardinality constraint is soft. In fact, the visualized portfolios in Figure 6 and Figure 8 (in the appendix) also "mostly" satisfy the constraint, where the portfolio weights are mainly concentrated on ~5 assets.
>
> We look forward to your feedback and we will be available for any of your further inquiries at your earliest convenience.
>
> **References**
>
> [1] Yoshua Bengio, Andrea Lodi, and Antoine Prouvost. Machine learning for combinatorial optimization: a methodological tour d’horizon. Eur. J. Operational Research, 290(2):405–421, 2021.
>
> [2] Nikolaos Karalias and Andreas Loukas. Erdos goes neural: an unsupervised learning framework for combinatorial optimization on graphs. In Neural Info. Process. Systems, 2020
>
> [3] T-J Chang, Nigel Meade, John E Beasley, and Yazid M Sharaiha. Heuristics for cardinality constrained portfolio optimization. Computers & Operations Research, 27(13):1271–1302, 2000

---

> ### Author Response · Authors · 2022-11-10
> **Response to Reviewer m4Wt (1/2)**
>
> Your thought-provoking comments and your thorough reading of our paper are deeply cherished. We regret some parts of this paper may cause certain confusion. Here we try to summarize the key concepts of this paper to present the general picture:
>
> **Our network is an end-to-end learning model for CO.** According to the taxonomy in Bengio's survey [1], our paper falls in line with the "end-to-end learning" paradigm of machine learning for combinatorial optimization (Sec 3.2.1 in [1]), whereby the problem definition is directly passed to an ML model, and the ML model directly outputs the solution, in a differentiable manner.
>
> **Our CO network is composed of a problem encoder (unconstrained) and our proposed optimal transport (OT) layer (to enforce constraints) for problem-solving.** The problem encoder network accepts "problem definition" and outputs $\mathbf{s}$. $\mathbf{s}$ is unconstrained and is further processed by our Gumbel-Sinkhorn OT layer to softly enforce the constraints, whose output is $\\{\widetilde{\mathbf{T}}_i | i=1,2,…,\\#G\\}$. There is index $i$ because there are $\\#G$ parallel Gumbel samples. Next, $\widetilde{\mathbf{T}}_i$ is treated as the decision variable for the CO cardinality constraint. Our theoretical study shows that the expected constraint violation of $\widetilde{\mathbf{T}}_i$ has a tighter upper bound than previous approaches.
>
> **Existing CO networks e.g. [2] do not enforce constraints in architecture.** Our OT cardinality layer is dropped in [2] and the output of problem encoder $\mathbf{s}$ is directly treated as the decision variable. Bounding the constraint violation via OT both theoretically and empirically is the main technical contribution of this paper, whereby the superiority is proved by our empirical study.
>
> **Comparing three portfolio optimization strategies.** For the portfolio optimization experiment, we would like to clarify among three strategies considered in this paper:
> 1. **History-opt.** This is the most basic setting of portfolio optimization that arises in most textbooks. It utilizes prices in the history to compute return $\mu$, risk $\Sigma$, and to formulate the optimization problem. The optimization problem could be tackled by off-the-shelf solvers. The investor follows the optimal portfolio in history and makes decisions, assuming that the return and risk characteristics will not change in the future. This strategy is named history-opt. Since history-opt does not involve any prediction model, its performance is inferior to prediction-based strategies.
> 2. **Pred-*then*-opt.** A direction of improvement is to predict the asset prices in the future (as accurately as possible) and then solve the optimization problem based on the predicted prices. It formulates a two-stage pipeline: 1) predicting prices by minimizing the prediction error, 2) solving the optimization problem. These two stages are unaware of each other. Since price prediction always contains errors, the optimization solution may be misled by the prediction error. In the experiment, pred-*then*-opt is better than history-opt, but there is still room for further improvement.
> 3. **Pred-*and*-opt.** Since the pred-*then*-opt pipeline does not consider the interference between the price predictor and the optimizer, in this paper, we propose a pred-*and*-opt pipeline by jointly training these two modules. Such a joint training pipeline is feasible because our proposed cardinality CO solver is differentiable. Pred-*and*-opt surpasses all competing methods in experiments.
>
> >***Q1&Q2: Some notations and concepts in the paper are unclear.***
> * We further elaborate on the notations/concepts you mentioned:
>     * The equations in Figure 1 could be further found in Sections 4 and 5. Under "Objective Estimator" are the equations used to estimate the objective score based on the transportation matrix from the OT layer. The equations are different to fit different problem formulations.
>     * $i,j$ are used for indexing, following conventions. In transportation matrix $\widetilde{\mathbf{T}}_i$, $i$ is the index of Gumbel samples.
>     * $\mu_{gt}$ and $\Sigma_{gt}$ are the mean return vector and the covariance risk matrix, respectively. And they are computed based on the ground truth prices, thus with the subscript "gt".
>     * We are using the "code-like" notations to picture the forward pass of our method neatly (they are exactly codes!).
>     * A problem encoder network plus our proposed OT cardinality layer compose a neural network that learns how to solve CO. We update Figure 1 with a blue box to highlight this concept.
>
>     To address your raised issues, the following revisions are made in PDF (marked as blue):
>     * We highlight in Figure 1 that the CO solving network is composed of a Problem Encoder and an OT Cardinality Layer.
>     * We elaborate on Page 2 and Page 9 about the unclear notations.

---

### Official Review · Reviewer_8ecY · 2022-10-24

**Confidence:** 3
**Correctness:** 4
**Technical Novelty And Significance:** 3
**Empirical Novelty And Significance:** 4
**Recommendation:** 6

**Clarity, Quality, Novelty And Reproducibility:**

The paper provides a novel, though somewhat incremental, method for handling top-k in combinatorial optimization networks. Instead of just optimal transport with entropy regularization smoothing solved via Sinkhorn algorithm, as in (Xie et al., NeurIPS'20), the authors add perturbation using Gumbel distribution. The authors show that their method, CardNN-GS, has lower theoretical bound on constrain violation than CardNN-S that relies on Xie et al.'s approach. They also show that the advantage holds empirically. The method outperforms the baseline CO approach, EGN by Karalias and Loukas, and performs on par with Gurobi. The paper is written clearly, though figures 3 & 4 are very small and hard to read.

**Strength And Weaknesses:**

The main strength of the paper is providing an extension to combinatorial optimization network framework that allows for bounding the constraint violation, for the case of cardinality constraints. This is achieved by incorporating SOFT top-k approach (Xie et al., NeurIPS'20) and adding a perturbation with Gumbel distribution. The paper is somewhat limited in its scope by the narrow focus on cardinality constraint - it would be stronger if it offered at least one simple example of how the approach can be extended to another constraint type.

**Summary Of The Paper:**

The paper provides an improved way for constructing combinatorial optimization networks for problems that involve cardinality constraints.

**Summary Of The Review:**

The paper is concerned with combinatorial optimization networks, a field of growing recent interest. It focuses on a single type of constraints - cardinality constraint - and introduces an improved method for handling it.

---

> ### Author Response · Authors · 2022-11-10
> **Response to Reviewer 8ecY (2/2)**
>
> >***Q2: Figures 3 & 4 are too small.***
>
> * In the revised PDF, we adjust the font sizes and marker sizes in Figures 3 & 4 and hopefully they are now easier to read.
>
> Please feel free to raise any further questions and we are looking forward to hearing your feedback!
>
> **References**
>
> [1] Zalan Borsos, Mojmır Mutny, and Andreas Krause. Coresets via bilevel optimization for continual learning and streaming. Neural Info. Process. Systems, 2020.
>
> [2] Abhimanyu Dubey, Moitreya Chatterjee, and Narendra Ahuja. Coreset-based neural network compression. In Eur. Conf. Comput. Vis., September 2018
>
> [3] Lingxiao Huang, Shaofeng Jiang, and Nisheeth Vishnoi. Coresets for clustering with fairness constraints. Neural Info. Process. Systems, 2019.
>
> [4] David Arthur and Sergei Vassilvitskii. K-means++: The advantages of careful seeding. ACM-SIAM Symposium on Discrete Algorithms, 2007.
>
> [5] Elias Khalil, Hanjun Dai, Yuyu Zhang, Bistra Dilkina, and Le Song. Learning combinatorial optimization algorithms over graphs. In Neural Info. Process. Systems, 2017.
>
> [6] O. Vinyals, M. Fortunato, and N. Jaitly, Pointer networks. Neural Info. Process. Systems, 2015.
>
> [7] Fudong Wang, Nan Xue, Jin-Gang Yu, and Gui-Song Xia. Zero-assignment constraint for graph matching with outliers. Comp. Vis. Pattern Recog., 2020.

---

> ### Author Response · Authors · 2022-11-10
> **Response to Reviewer 8ecY (1/2)**
>
> Many thanks for recognizing our novelty, as well as for the constructive comments and suggestions for our paper. We set out below our response to your questions:
> >***Q1: The scope of cardinality constraint may be limited. It would be stronger if it offered at least one simple example of how the approach can be extended to another constraint type.***
>
> * For the concern that the scope of this paper is limited, we try to clarify it as follows.
>
>   **Cardinality, especially hard enforcement of cardinality is widely practical and useful.** The cardinality constraint itself is a well-established area with a long-standing and broad impact on machine learning, theoretical computer science, operation research, and finance (just to name a few).
>
>   For example, the hard cardinality constraint can be treated as a special case of the coreset problem which has been intensively studied in recent machine learning and vision conferences [1,2,3] which is in fact currently towards the between of theoretical computer science [4] and machine learning. Our work can also handle soft cardinality (i.e. the general coreset) because the Gumbel-Sinkhorn is a continuous relaxation.
>
>   Given the fact that existing cardinality constraint solvers are mostly learning-free, we strongly believe our neural solver with bounded constraint violation can be of great value to both academia and industry whereby the cardinality constraint needs to be satisfied.
>
>   **The proposed paradigm can incorporate other constraints beyond cardinality.** As stated above, we originally think cardinality itself is of great importance to deserve a full research paper. While in fact, our end-to-end one-shot CO learning and solving paradigm could further incorporate other types of constraints.
>
>   When preparing this conference paper, our intention was not to distract the theme of this paper to other problems so we decided to focus on cardinality. Yet it is indeed feasible to extend our work to other types of constraints. We add the permutation constraint as an example here as encoded by Sinkhorn and optimal transport (OT). We are a bit refrained from elaborating on this point in the main paper as it needs to involve more comparison and discussion on the large body of permutation optimization problems and approaches. But we would be happy to add more details and discussions if the reviewers felt that they are needed.
>
>   The permutation constraint can be directly applied to popular problem instances like TSP and graph matching under our one-shot learning and problem-solving paradigm. Compared with the mainstream auto-regressive scheme in learning for solving TSP (e.g. S2V-DQN [5] and pointer net [6]), conceptually, our one-shot model can be more efficient. We report our preliminary results on using our paradigm for solving TSP on synthetic graph data of size 20, 50, and 100 as the mean of 10000 trials as follows:
>
>   | optimal gap | TSP20 | TSP50  | TSP100 |
>   | - | - | - | - |
>   | Sinkhorn | 0.27% | 0.95%  | 2.48% |
>   | S2V-DQN [5] | 1.42% | 5.16%  | 7.03%  |
>   | pointer-net [6] | 1.15% | 34.48% | - |
>
>   One step further, we are also exploring a unified Sinkhorn network to tackle both permutation and cardinality constraints, whereby its application could be found in graph matching with outliers in both graphs. Our preliminary result on Pascal VOC dataset (with outliers) is as follows:
>   | graph matching with outliers | accuracy |
>   | - | - |
>   | Sinkhorn net for permutation+cardinality | 60.4 |
>   | Sinkhorn net for permutation | 58.8 |
>   | ZACR solver [7]  | 30.2 |
>
>   The above results are preliminary proof-of-concepts, and we believe there is room for further improvement. If you feel needed, we are happy to elaborate on the above results and technical details in the main paper (may need to reorganize the paper due to space limitations).

---

### Author Response · Authors · 2022-11-23
**Inquiry for post-rebuttal comments**

Dear AC and reviewers,

We truly appreciate your valuable efforts for the ICLR community and your thought-provoking comments on this paper. We also appreciate Reviewer m4Wt for the in-depth discussion with us and for raising her/his score. Since the author-reviewer discussion period is approaching its end, we would be eager to know: Have our rebuttal and our revised PDF addressed your concerns?

We are also always available to respond to any of your further questions.

Best,

Authors of Paper 760

---

### Decision · Program_Chairs · 2023-01-20

**Decision:**

Accept: poster

**Justification For Why Not Higher Score:**

The paper is clearly on the fence, borderline.

**Justification For Why Not Lower Score:**

Although I find the main idea of this work original (the concept of integrating soft top-k in a binary selection problem), and I think Figure 3 in particular is interesting, this paper could be rejected because of poor presentation standards and an experimental section in 4 that's questionable (using financial data for portfolio selection) given the low interest of the ICLR community for these problems.

**Metareview: Summary, Strengths And Weaknesses:**

This paper proposes a specific relaxation of "combinatorial" solvers. Instead, what is mostly described here is a solver for binary optimization.

The solver is proposed to solver binary problems in an amortized way (and compared with, e.g. gurobi) in Section 3, but also to handle end-to-end learning problems, in which the representation/prediction for data are learned in the inner loop, on a portfolio selection task in Section 4.

The paper suffers from a few issues (writing is not clear and clumsy in several places, cosmetically the paper is quite ugly, e.g. Fig. 1 provided with a low res definition and tiny fonts, the title is ridiculously long and not so informative, what paper claims to consist in "notes"??) and the application to portfolio selection is not exciting (validation on financial is always rife with issue, where the signal-to-noise ratio is usually very low; there's low interest of the ICLR community for these problems).

). In particular, one wonders how much of hyperparameter tuning was needed to reach these performances, this is not mentioned in the main body of the paper.

The paper has received a mild appreciation from reviewers, but no enthusiasm. I do, however, suggest acceptance as a poster, because the experiments are substantial, notably Fig. 3 and I find the idea overall interesting. If accepted, I expect the authors to put a significant amount of work into improving presentation.

**Note From Pc:**

if the above contains the word "oral" or "spotlight" please see: "oral" presentation means -> notable-top-5% and "spotlight" means -> notable-top-25%. As stated in our emails, we are disassociating presentation type from AC recommendations